# Recombination junctions from antibody isotype switching classify immune and DNA repair dysfunction

Clara Vázquez García [1,2,14], Benedikt Obermayer [3,14], Baerbel Keller[4,5], Mikhail Lebedin [1,2], Christoph Ratswohl [1], Hassan Abolhassani [6,7], Antonia Busse[1,2,8], Michela Di Virgilio [1,2], Stephan Mathas[1,2,9], Dorothee Speiser[2,10], Dieter Beule[3], Qiang Pan-Hammarström [6], Klaus Warnatz [4,5] & Kathrin de la Rosa [1,11,12,13] ✉

Personalized assessment of immunocompetence and DNA double-strand break (DSB) repair requires methods that are sensitive to genetic and molecular complexity beyond the well-known monogenic disorders. Inspired by decades of research using B cells to study DNA repair processes, here we present SWIBRID (SWItch junction Breakpoint Repertoire IDentification), a tool to systematically profile genomic junctions generated in vivo during antibody class switch recombination (CSR) in B cells. As CSR junctions reflect immune diversity and DNA repair proficiency, SWIBRID detects phenotypic manifestations of deficiencies via a highly scalable, blood-based PCR followed by long-read sequencing and bioinformatic analysis. We show that specific DNA repair defects, including cancer-associated mutations, exhibit distinct CSR junction patterns. Notably, SWIBRID distinguishes different types of DSB repair knockouts and identifies the respective genetic defect in cell lines. In 68 patients, we detect immunodeficiencies and DNA repair defects with high accuracy (area under the curve 0.99 and 0.84, respectively), and identify previously uncharacterized patient groups as well as patient-specific CSR junction signatures. With SWIBRID, we seek to advance the identification of pathogenic defects, support early diagnosis, and address molecular heterogeneity that drives variable clinical outcomes.

Adaptive immune responses rely on a diverse repertoire of immune cells generated by somatic diversification of antigen receptors[1]. The underlying molecular processes include the programmed induction and repair of DNA double-strand breaks (DSB), which involves a complex network of repair pathways[2,3]. Defects in such pathways can negatively impact the adaptive immune system and jeopardize genome integrity, potentially leading to cancer, radiation sensitivity, premature ageing, neurodegenerative diseases, or developmental disorders[4–9]. To this end, numerous technical approaches have been developed to uncover defects in immunity or DSB repair, and to inform clinical decisions for treatment and prevention.

Newborn screening tests such as T-cell receptor excision circles (TREC) and κ-deleting recombination excision circles (KREC) have revolutionized the early identification of inborn errors of immunity (IEI) related to T and B cell defects[10–12]. However, these assays can fail to identify defective antibody responses as observed in common variable immunodeficiency (CVID)[13,14].

Genetic testing identifies monogenic defects predisposing to cancer and impacting its treatment[15,16]. Beyond cancer, such testing can also benefit patients with immunodeficiencies, offering targeted treatment options. For instance, gene therapy has emerged as a promising treatment for defects in the DNA repair enzyme Artemis[17].

Given that single, high-risk mutations are infrequent in the population while multifactorial diseases are often driven by many combined low-risk variants[18–21], methods assessing deficiencies in single genes or proteins are unlikely to reflect the overall performance of the immune system and the DNA repair landscape. Therefore, more holistic approaches have addressed DNA repair performance in primary samples by directly measuring repair outcome, for instance, analyzing the quality and quantity of genomic alterations in whole genome sequencing (WGS) data[22,23]. While such methods avoid indirect in vitro assays, WGS data comes with considerable costs and technical challenges, as well as data privacy concerns. Alternative methods to assess immune health are based on high-dimensional phenotypic readouts such as transcriptomics and proteomics[24].

In contrast, we reasoned that deep characterization of natural DSB hotspots in B cells holds the potential for a targeted, yet highly versatile solution to assess the quality of DNA repair and immunocompetence. Namely, the intron of the Immunoglobulin (Ig) heavy chain (IGH) locus accumulates DSB repair events in switch (S) regions. Such events support the diversification of the antibody effector repertoire in a process called class switch recombination (CSR)[25,26]. B cells have therefore been used for decades to study DNA repair phenotypes of specific genetic knockouts (KO)[27–31].

During CSR, B cells rearrange the IGH locus to change the isotype of the resulting antibody from IgM to IgG, IgA, or IgE[32]. The process results in the replacement of the Cμ constant region exons with the Cγ, α, or ε exons located 100 to 200 kb downstream. S regions are GC-rich, repetitive intronic sequences located upstream of C exons encoding for constant domains. S regions are recombined with each other upon introduction of DNA lesions by activation-induced cytidine deaminase (AID). Lesions are converted into DSBs and repaired by several distinct pathways, resulting in CSR junctions. The dominant DNA repair pathway active during CSR is nonhomologous end joining (NHEJ), followed by microhomology-mediated end-joining (MMEJ), which is linked to aberrant repair and chromosomal translocations[31,33–35]. CSR junctions resulting from NHEJ are characterized by direct joining or linked to microhomologies of a few base pairs, while MMEJ is associated with longer stretches of homology, resections, and deletions[36]. Homologous recombination (HR) repairs DSBs with high accuracy, restoring the original sequence, and is considered mostly dispensable for CSR[37]. However, deficiencies in HR repair factors, such as BRCA1 (Breast cancer type 1 susceptibility protein), have been found to alter the sequence patterns surrounding CSR junctions in human[38].

Here, we hypothesize that the repertoire of CSR junctions in class-switched memory B cells captures the diversity of antibody effector components and the performance of diverse DNA repair pathways. We introduce SWIBRID (SWItch junction Breakpoint Repertoire IDentification), an approach that systematically characterizes in vivo generated CSR junctions with the aim of aiding DNA repair research and improving the identification of immunocompromised individuals.

## Results

### A methodology to systematically profile class switch recombination (CSR) junctions

To evaluate isolated CSR junctions of different isotypes in a single reaction, we optimized a PCR amplification using 4 barcoded primers. In the human IGH locus, the forward (FW) primer binds 565 bp upstream of the human Sμ region[39] while reverse (RV) primers for Sγ-, Sα-, and Sε bind 775 bp (γ3), 775 bp (γ1), 63 (γ2), 777 bp (γ4), 706 (α1), 169 (α2), and 161 bp (ε) downstream of the respective S regions (Fig. 1a). FW and RV primer binding sites are multiple kilobases apart in the genome, ensuring that primers exclusively amplify IGH genes from switched memory B cells. The number of PCR cycles (25) was kept as low as possible to minimize technical noise while still ensuring sufficient representation of the biological diversity. We subjected amplicons from blood-derived B cells to MinION long-read sequencing and

designed a dedicated bioinformatics pipeline (see "Methods", Supplementary Fig. 1a). Amplicons ranged from 1000 to 4000 bp in size (Supplementary Fig. 1b–e). Reads meeting our quality criteria were mapped to the genome and hierarchically clustered using similarity between individual alignment patterns. To assess and suppress technical noise likely caused by long-read sequencing and mapping artefacts, we analyzed reads from a linearized Sμ region-containing plasmid and subsequently ignored small indels for clustering (Supplementary Note 1 and Supplementary Fig. 1f, g). We performed extensive benchmarking experiments on simulated data to optimize our method with regard to a clustering cutoff value and a downstream cluster filtering strategy (Supplementary Note 1). We next applied our pipeline to human B cell samples with defined clonality. As expected, analysis of an in-house monoclonal cell line revealed two main CSR clones in accordance with 2 switched alleles (Fig. 1b, c). We observed some heterogeneity in the monoclonal cell line, where subtle sequence variations around the breakpoint (Supplementary Fig. 1h) formed independent clusters in the read plot, leading to a slight overestimation of diversity for this sample. PCR may introduce such sequence variation, especially in highly repetitive DNA, but a similar level of heterogeneity in both PCR-amplified samples and the non-PCR plasmid control (Supplementary Fig. 1g) suggests that most of the variation arises from clustering, mapping, and sequencing errors. An oligoclonal line showed several switched alleles, and a polyclonal sample of primary human B cells displayed a high and evenly distributed diversity. We also adapted our primer design, PCR protocol, and bioinformatical pipeline to analyze mouse CSR junctions and obtained amplicons from 2500 to 4000 bp in size (Supplementary Fig. 1i, j). We used the mouse CH12 cell line and splenocytes to validate our assay (Supplementary Fig. 1k, l).

To further confirm an adequate quantification of CSR clones and understand the limitations of the pipeline, we assessed different numbers of (i) in silico generated CSR junctions representing diverse clonotypes and (ii) primary B cells to address the impact of the input cell number on output SWIBRID clusters. First, we analyzed simulated reads generated using the coordinates of CSR junctions derived from B cell sequencing data (31 human donors). The sequences were further diversified by adding mutations, insertions, and deletions according to the observed technical noise characteristics (see "Methods"). As long as the input read number was sufficiently large, the number of output clusters correlated strongly with the input clonality (Fig. 1d; $R^2 = 0.98\text{-}1$ for 10,000 to 100,000 reads). Second, we tested the effect of distinct B cell numbers using class-switched memory B cells of three independent donors in triplicates. When using a maximum of 50,000 reads of increasing amounts of switched memory B cells, a solid correlation was observed (Fig. 1e). A saturation of the resulting diversity was noticed beyond an input of 20,000 sorted B cells. We therefore used no more than 200,000 peripheral blood mononuclear cells (PBMC) per sample for further analysis, typically containing 5–10% total B cells and around 10,000 switched B cells. To benchmark the reproducibility of our assay, we devised a meta-clustering strategy to trace recurring clones across donors (Supplementary Note 2 and Supplementary Data 1). We observed matching clusters much more often in sample pairs from the same donor than from different donors. A baseline number of shared clusters between different donors likely reflects both the presence of public clones and a limited resolution of our approach (Fig. 1f). To compare SWIBRID with the standard approach for B cell analysis, we evaluated whether our genomic findings are reflected at the RNA level. Thus, we analyzed IGHA and IGHG isotype frequencies between CSR junction alleles and BCR mRNAs from 50,000 to 100,000 circulating B cells from three donors. Both approaches obtained similar outcomes regarding the percentage of IGHA (Fig. 1g). In conclusion, we have established a high-throughput pipeline to profile and quantify the diversity of CSR-derived clones from human blood samples.

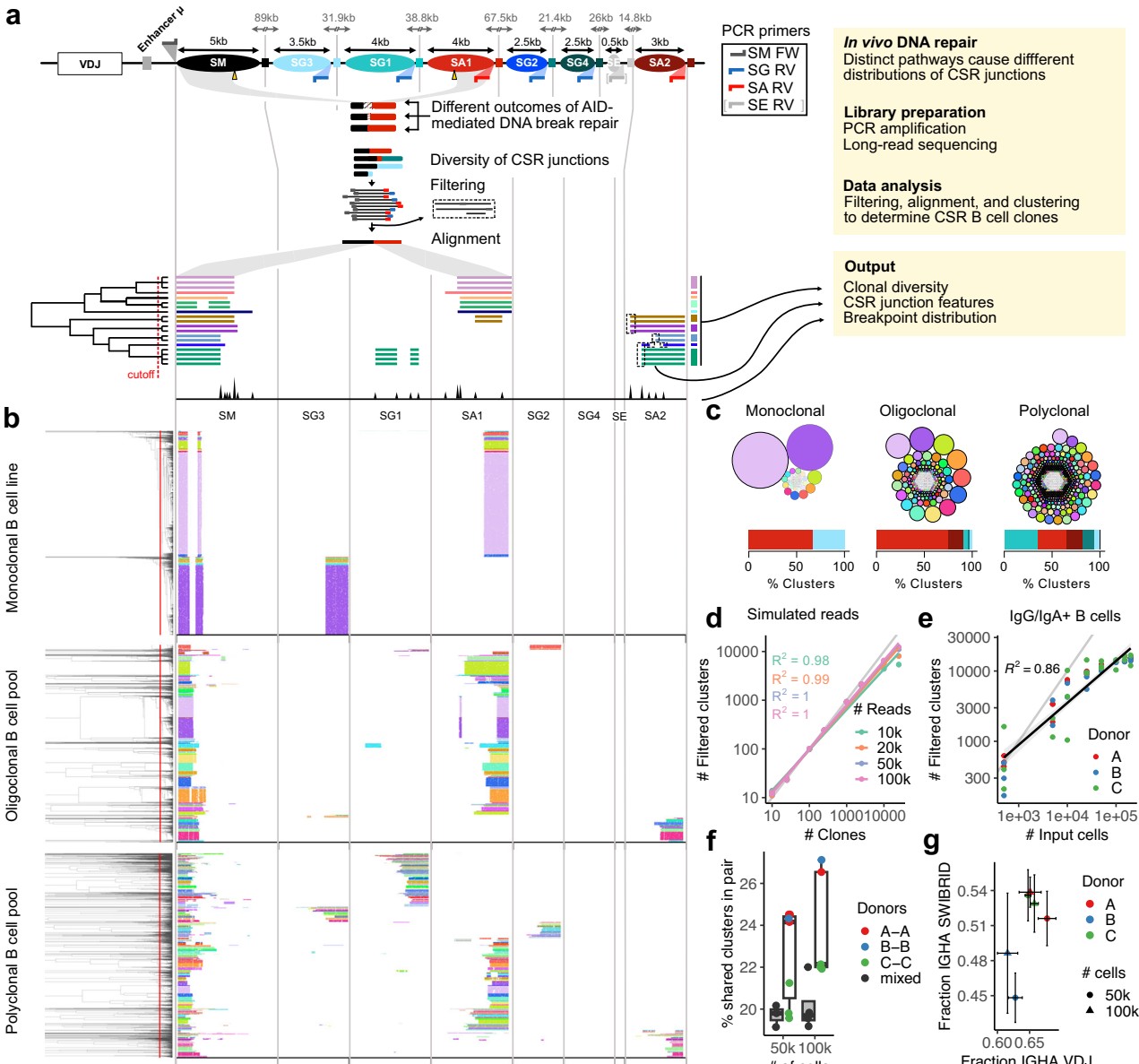

**Fig. 1 | Clustering of long reads covering class switch recombination (CSR) junctions enables quantification of B cell clones. a** Scheme of the SWIBRID pipeline. Top: the human immunoglobulin heavy chain with switch (S) regions as oval shapes. Sizes and distances between S regions are indicated in kilobases (kb). Middle: the amplified junctions reflect the diversity generated during CSR; sequenced reads are aligned to the genome and clustered by cutting the dendrogram at a cutoff indicated by the dotted red line; clusters are shown in the same color and considered unique, switched alleles. Bottom: histogram of breakpoint positions. Right: the output of the pipeline consists of features derived from clustered reads and breakpoint junctions. **b** SWIBRID was performed with 2500 reads of B cell samples with indicated clonality: (i) a monoclonal B cell line obtained from EBV immortalization containing two switched alleles; (ii) an oligoclonal, EBV-immortalized B cell pool; (iii) a polyclonal B cell pool prepared from a sample of 25,000 sorted IgG/IgA[+] B cells. The colour of clones is arbitrary. **c** Plots depicting

clones from the samples in (**b**). Every circle represents a clone with a unique CSR junction. The size of the circle corresponds to the number of reads. Colours are matched with (**b**). A stacked bar plot shows the percentage of distinct isotypes with colours matching the respective S regions depicted in (**a**). **d** Filtered clusters determined by SWIBRID using indicated numbers of simulated input clones and reads. $k = 1000$. $R^2$ = coefficient of determination from the linear regression. **e** Filtered clusters obtained as function of input numbers of IgG/A[+] B cells of three donors, together with regression line (shaded area indicates 95% confidence interval). **f** Percentage of shared clusters in independent samples of the same donor (A-A, B-B, C-C) or different ones (mixed) at 50,000 and 100,000 B cells ($n = 3$; boxes indicate 25th to 75th percentile; whiskers extend to largest/smallest value no further than 1.5× interquartile range, lines indicate median). **g** IgA isotype frequencies determined via genomic CSR junctions (SWIBRID) versus B cell repertoire analysis based on BCR transcripts (error bars denote s.e.m.; $n = 3$ donors).

## Gathering CSR junction characteristics reflective of immunocompetence and DNA repair in human and mouse

To deeply profile the CSR junctions as a result of DSB repair and diversification, we established a strategy to condense a full dataset into a vector of 68 features related to 5 main categories (Figs. 1a and 2a and Supplementary Fig. 2a–e): (i) structural aberrations ($n = 5$) such as duplications, templated inserts, and the frequency and size of

inversions, which we define as stretches within S regions with reversed orientation (Supplementary Fig. 2d); (ii) sequence context ($n = 17$) such as microhomologies, untemplated nucleotides, and sequence motifs directly adjacent to the junctions (Supplementary Fig. 2a, b, e); (iii) the breakpoint matrix ($n = 28$) of all donor- and acceptor S regions to quantify frequencies of breaks within or between regions, as well as the dispersion of breakpoint locations, the fragment length, and GC

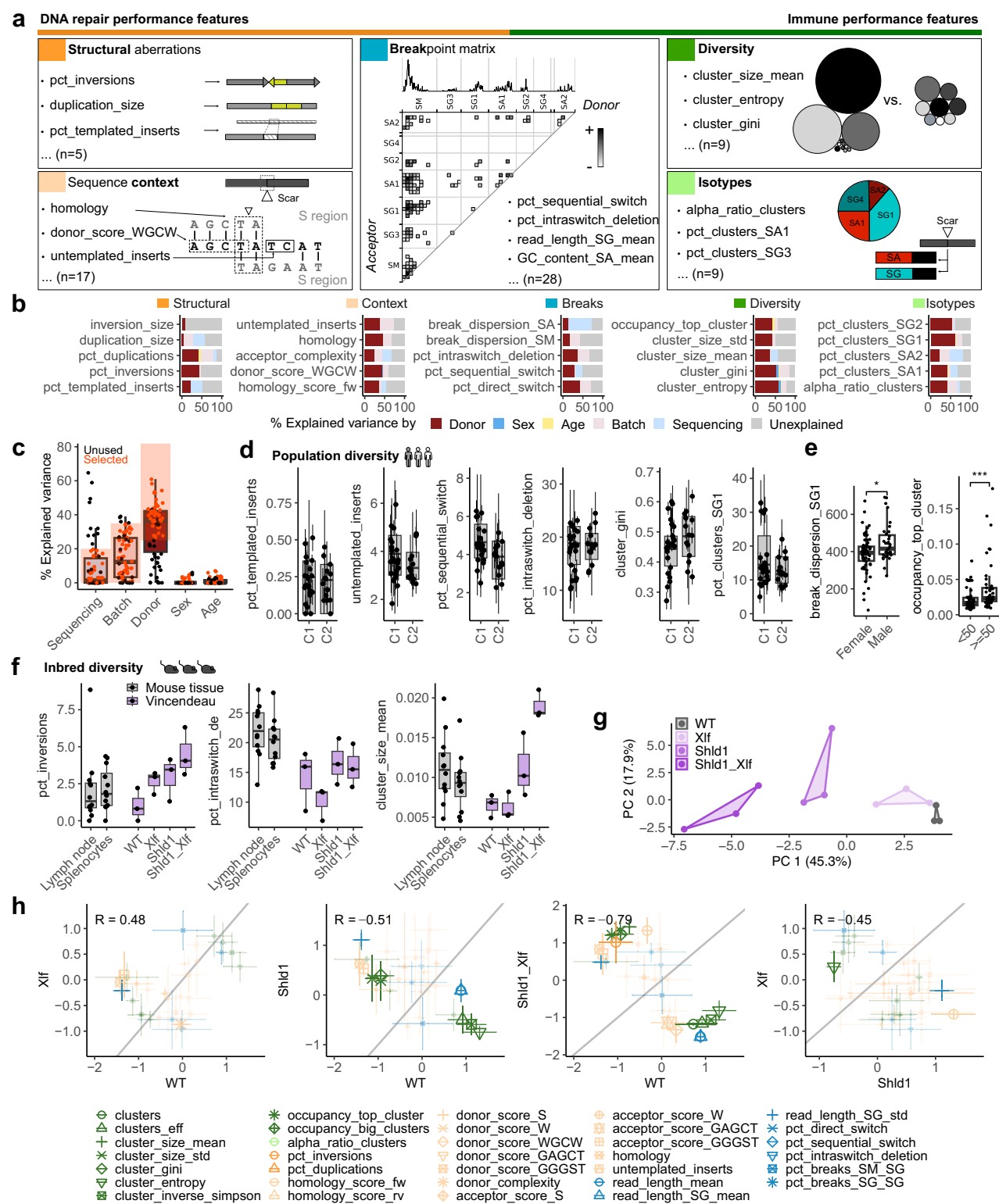

content (Supplementary Fig. 2c); (iv) diversity measures ($n = 9$) such as cluster number and mean size, and the entropy and other statistics of the size distribution; and (v) isotype composition ($n = 9$). While groups (i) and (ii) are expected to reflect the performance of distinct DNA repair pathways, group (iii) represents a combined measure of DNA repair pathway engagement and immune performance. Groups (iv) and (v) should mirror the capacity of the immune system to respond to threats. We evaluated these features on a large dataset of 98 samples from 40 donors across 7 batches (cohort C1). A batch of 10 samples

was sequenced with MinION as well as PacBio HiFi technology, giving very similar results (Supplementary Note 2 and Supplementary Data 1). To assess the robustness of our features across different batches or sequencing methods, and their ability to discriminate between donors, we used variance partitioning analysis to verify that for most features, the bulk of the variability is explained by donor identity (Fig. 2b, c and Supplementary Fig. 2f). This indicates that our quantification yields accurate and reliable output. An independently collected and processed set of random human blood donors (cohort C2) with 41 samples

**Fig. 2 | A catalogue of CSR junction characteristics to assess immunocompetence and DNA repair in human and mouse. a** Overview of $n = 16$ representative of $n = 68$ total SWIBRID features, grouped into 5 indicated categories. Small schemes were used to depict the respective features. Breakpoint matrix features include a histogram of breakpoints occurring between the donor S region ($x$-axis) and the acceptor S region ($y$-axis) with rectangles and darkness denoting breaks and their frequency, respectively. **b** Percentage of variance explained by donor identity, batch, or sequencing method for selected features from each category. **c** Box plots of explained variance for all features. "Robust" features with > 30% variance explained by donor, <20% by sequencing, and <25% by batch were selected for downstream analyses. **d** Range of selected features in two different cohorts of healthy donors (C1: 98 samples of 40 donors; C2: 41 samples of 14 donors). **e** Left: break_dispersion_SG1 values (female, $n = 20$; male, $n = 20$). Right:

occupancy_top_clone values (age < 50, $n = 21$; age ≥ 50, $n = 19$). $P$-value from two-sided Wilcoxon test (*: $p < 0.05$, ***: $p < 0.001$). **f** Box plots of representative robust features generated with splenocytes and lymph node cells of immunized mice (grey, $n = 12$), combined with a re-analysis of CSR long-read sequencing data from Vincendeau et al.[30] (violet, $n = 3$). **g** PCA of re-analyzed Vincendeau data[30] using 32 of 44 robust features, excluding features related to isotypes missing in the dataset and features missing in mouse compared to human. **h** Comparison of scaled mean features of genotypes (Vincendeau data). Significantly different features are indicated using larger symbols (adj. $p$-value < 0.05 from two-sided $t$-test, with Benjamini–Hochberg correction). Boxes in (**d** and **f**) indicate 25th to 75th percentile; whiskers extend to largest/smallest value no further than 1.5× interquartile range, lines indicate median.

from 14 individuals (6 batches) showed very similar distributions of features across the donor sets (Fig. 2d). For all further analyses in this manuscript, we selected a smaller set of 44 robust features (Supplementary Data 2) with a high percentage (> 25%) of variance explained by donor identity and not primarily attributed to technical batch effects or the sequencing method (< 35% and < 20%, respectively; Fig. 2c). Of note, most features were not strongly correlated with age or sex. As exceptions and in line with previous literature, we found that the size of the biggest clone tended to be higher in older donors (≥ 50 y)[40], while the breakpoint dispersion in Sγ was slightly lower in females (Fig. 2e). We additionally attempted to call single-nucleotide variants in our data. We classified as likely somatic those that were not annotated in dbSNP (database of Single Nucleotide Polymorphisms), but had intermediate allele frequencies and did not segregate across clusters as would be expected for heterozygous germline variants (see "Methods"). However, features related to variants were the least robust in discriminating between donors and the most affected by sequencing technology (Supplementary Fig. 2f). In contrast, the selected set of robust features showed a high correlation across sequencing methods (Supplementary Fig. 2g). Since some of our features measure similar properties of the data in different ways (e.g., diversity), we assessed correlations between features over the C2 cohort. As expected, we found groups of highly correlated features, but also a strong association of isotype frequencies with multiple features related to length, GC content, homology, or motif occurrences (Supplementary Fig. 2h). However, calculating context features separately for each isotype rather than globally did not produce more robust outcomes. We also confirmed increased robustness by calculating diversity features on subsets of 1000 reads and by averaging features first within clusters and then over the entire dataset (Supplementary Fig. 2i).

Beyond human, we used samples from spleens and lymph nodes of inbred immunized mice to demonstrate that our assay can be used in different mouse tissues with comparable results (Fig. 2f). To further benchmark the detection of CSR-related signatures of defective DNA repair, we re-analyzed a public dataset of Sμ-Sγ1 junction reads from splenic B cells of specific-pathogen-free mice generated by Vincendeau et al.[30] (see "Methods"). This study investigated the role of Shieldin 1 (Shld1) in preserving end-protection of AID-mediated DSB by restricting resection in NHEJ-proficient and NHEJ-deficient ($Xlf^{-/-}$) B cells. We recognized that many features cannot easily be compared between human and mouse data, or between data sets with different primer designs (Supplementary Fig. 2j, Fig. 1a, and Supplementary Fig. 1i). Still, within the dataset of Vincendeau et al., we observed strong differences between different genotypes and not only for features directly connected to resection as observed in their work[30] (Fig. 2f and Supplementary Fig. 2j). Moreover, in a principal component analysis (PCA) of that dataset, all KO samples separated well from WT controls and from each other, while $Xlf^{-/-}$ was most similar to WT and $Shld1^{-/-}Xlf^{-/-}$ most dissimilar (Fig. 2g). We next created a scatter plot of standardized mean features to compare features between pairs of genotypes and highlight the most distinctive ones (Fig. 2h). Specifically, SWIBRID confirmed a

hyper-resection phenotype for the $Shld1^{-/-}Xlf^{-/-}$ double KO, as features that depend on read length and break dispersion along the Sγ1 region in CSR junctions were significantly altered. Compared to $Xlf^{-/-}$, $Shld1^{-/-}$ strongly impacted diversity features such as cluster distribution, size, and entropy, in accordance with strongly decreased serum levels of IgG and significantly reduced frequencies of IgG class-switched B cells in these mice[30]. We also identified additional genotype-specific features, such as the expansion of single clones in the repertoire (occupancy_big_clusters) and a lower GC content in the acceptor side of the break (acceptor_score_W) for the KO of $Shld1$ in the presence or absence of $Xlf$.

Collectively, we show that SWIBRID delivers a reproducible and informative quantification of CSR junction characteristics that can be applied in diverse B cell-containing tissues to deeply explore the contribution of genetic factors on DSB repair and immune diversity in a single analysis.

## CSR junction footprints serve for genotype prediction of DNA repair knockouts

Next, we aimed to better map the landscape of DSB repair at CSR junctions. We applied SWIBRID to CH12 mouse cell lines carrying monogenic KOs of further DNA repair factors, namely (i) $Lig4$ (DNA Ligase IV), which joins DSBs during NHEJ[36], (ii) $Trp53bp1$ (P53-binding protein 1), which acts as a platform to protect DNA ends from resection by recruiting pro-NHEJ factors upon Ataxia Telangiectasia Mutated (ATM) kinase phosphorylation[41], (iii) $Rif1$ (Rap1-interacting factor 1), which partners with $Trp53bp1$ to promote NHEJ[42], and (iv) $Brca1$, which plays a key role in precise HR repair and competes for the DNA break site with $Trp53bp1$ and $Rif1$[43]. CH12 cells were stimulated for two days with a cocktail of interleukin-4, anti-CD40 antibody, and TGFβ1, which induces CSR from IgM to IgA in vitro[44] (Fig. 3a). A PCA combining the data from CH12 with spleen and lymph node samples from Fig. 2f revealed that the cell lines were not directly comparable to primary mouse samples that underwent CSR in vivo (Fig. 3b). The difference was only partly explained by the restriction of CH12 to IgA (Fig. 3b). Notably, splenocyte and lymph node samples were more variable and globally more distinct from CH12 samples with diverse KOs. However, with few exceptions, the CH12 samples clustered according to their genotype in a heatmap of repair outcomes represented by $n = 32$ features (of the 44 robust features defined in a human dataset in Fig. 2, we excluded features missing in the mouse cell lines, such as features related to Sγ, isotype frequencies, and sequential breaks; Fig. 3c and Supplementary Data 2). For every genotype, we could identify features that were significantly different from WT samples (Fig. 3d). For instance, the direct switching from Sμ to any other S region ("pct_direct_switch") was significantly higher for $Brca1$ KO compared to WT, and the mean length of PCR products, inversely reflective of resection, was significantly lower for $Trp53bp1$ and $Lig4$ KO and higher in $Brca1$ KO. The variability of CSR junction positions across the Sα region ("break_dispersion_SA") was significantly increased in $Trp53bp1$ KO and $Lig4$ KO, while reduced in $Brca1$ KO. As previously reported[45],

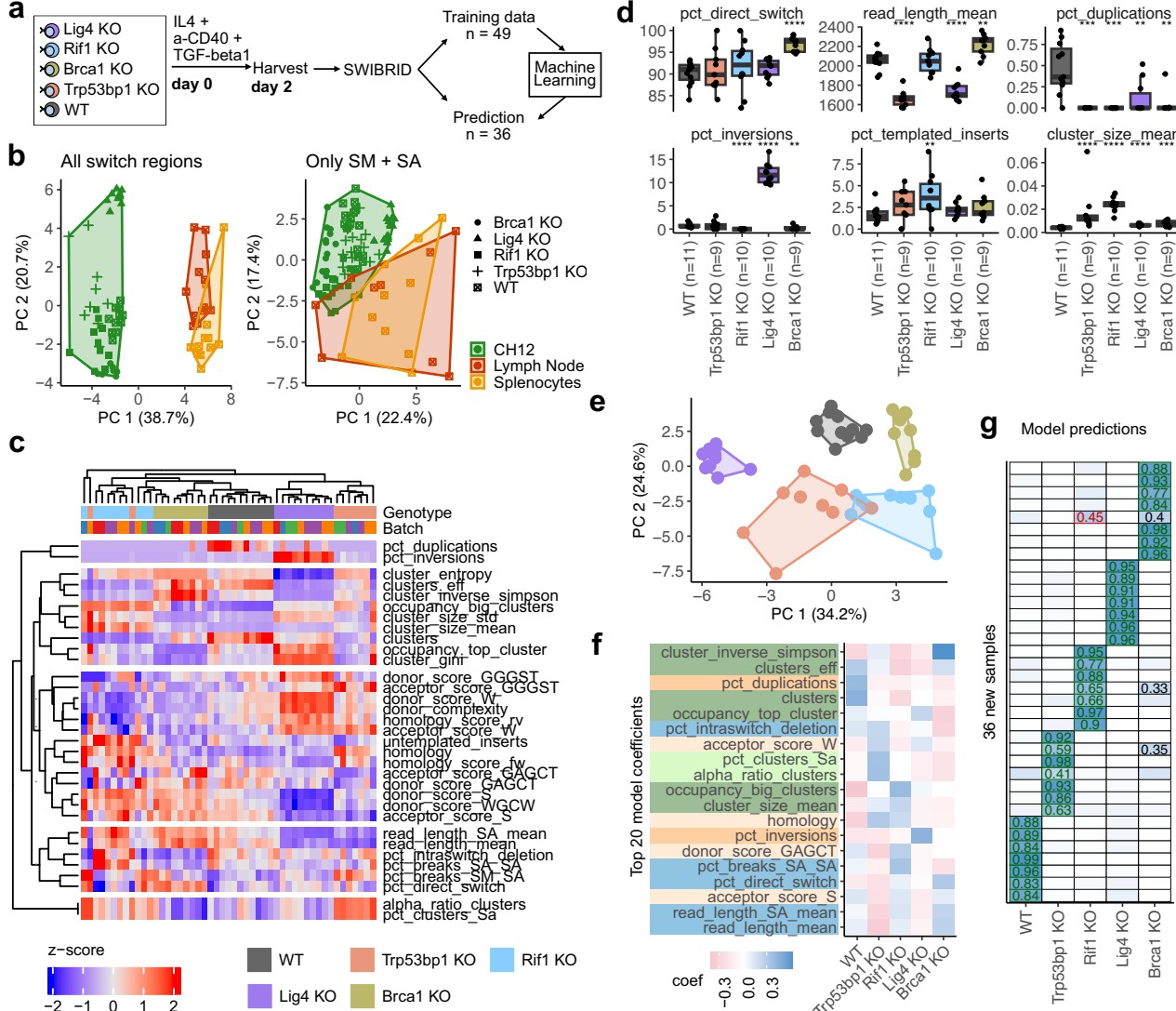

**Fig. 3 | Knockouts of DNA repair proteins in CH12 cells show specific DNA junction footprints that serve for genotype prediction. a** Scheme of CH12 in vitro activation. IL4 interleukin-4, a-CD40 anti-mouse CD40 antibody, TGFβ1 Transforming Growth Factor Beta 1. **b** Principal component analysis (PCA) of WT CH12 cells, CH12 cells with knockouts (KO) of Brca1, Lig4, Rif1, and Trp53bp1, as well as splenocytes, and lymph nodes from WT mice using robust SWIBRID features. Left: including reads from all switch regions (42 of 44 robust features−excluding features missing in mice compared to human). Right: selecting reads with Sα primer (33 of 44 robust features−excluding Sγ and human-specific features). **c** Heatmap of row-scaled features of CH12 WT and knockouts (n = 49 samples). Sample genotypes and batches are indicated on top. **d** Box plot of representative

features. *P*-values from two-sided Wilcoxon test compared to WT (****: *p* < 0.0001, ***: *p* < 0.001, **: *p* < 0.01, *: *p* < 0.05; boxes indicate 25th to 75th percentile; whiskers extend to largest/smallest value no further than 1.5× interquartile range, lines indicate median) **e** PCA using 33 robust features for CH12 cells. Genotypes are colored as in (**c**). **f** Top 20 coefficients in a multinomial ridge regression model trained on the 49 training samples. Features are colored according to the classification of Fig. 2a. **g** Regression results for the genotype prediction of 36 independent CH12 samples done with a machine learning model using 33 robust features of 49 CH12 samples. Prediction scores for each genotype are indicated by background colors (white = low, blue = high) and values > 0.25 are shown. Green: correct predictions, red: incorrect predictions.

CSR breaks leading to inversions of S regions were found to be substantially higher in *Lig4* KO but decreased for *Rif1* and *Brca1* KO. Frequencies of templated insertions[46] were highest for *Rif1* KO. As expected, mean cluster size for all KOs was significantly increased, reflecting a reduced ability of B cells to diversify. All genotypes separated well in a PCA, despite some overlap of *Rif1* KO with *Trp53bp1* KO samples (Fig. 3e), which may be expected as they interact to protect DSBs against nucleolytic processing[47,48].

Next, we re-analyzed published CSR junction data from mouse CH12 cells and splenocytes with KOs in various DSB repair proteins, including *Lig4* and *Trp53bp1*[27]. Using our bioinformatics pipeline on data generated by linear amplification and short-read sequencing, we recapitulated previous findings such as reduced mean length or an increased

use of >6 nt homology in the *Trp53bp1* mutant (Supplementary Fig. 3a−d). Additionally, we identified a higher frequency of intra-switch inversions in *Lig4* KO, which was even more pronounced in *Trp53bp1* KO (Supplementary Fig. 3c). We also observed a striking decrease in GC content at the acceptor side of CSR junctions for both KOs. As expected, the bioinformatics pipeline of SWIBRID showed improved performance for breakpoint matrix features when long reads were available. Nonetheless, the assessment of sequence context features like 1−3 nt microhomology or blunt ends benefits from sequencing techniques with low error rates and experimental designs that preferentially amplify CSR junctions from high-complexity regions.

Finally, we applied machine learning based on a multinomial ridge regression model trained on 49 samples to predict defects in a newly

generated set of 36 additional CH12 samples ("Methods"). Notably, the correct genotype was predicted for 35 out of 36 independent samples (for 32 of them with high confidence; Fig. 3f, g). Only one *Brca1* KO sample was wrongly assigned to a *Rif1* KO, showing *Brca1* KO only as a second possible hit. Cross-validation using either one batch or 25% of the samples for testing and the rest for training, resulted in 96% and 94% prediction accuracy, respectively.

Taken together, SWIBRID confirms previously reported impacts of specific DSB repair KOs on CSR phenotypes. Moreover, our attempt to globally define CSR junction characteristics identified novel features, such as donor/acceptor sequence motifs and templated insertions. Importantly, our data suggest that genetic defects affecting different pathways of DSB repair can be identified by SWIBRID.

### Identification and classification of CVID patients by SWIBRID

No single clinical feature or laboratory test can definitively confirm the diagnosis of CVID[18,49,50]. We therefore examined whether SWIBRID could reliably identify CVID patients. We obtained two to three technical replicates each from 21 patients with CVID or related diagnoses and 14 controls for analysis by SWIBRID ("CVID training" dataset; Fig. 4a). As expected, the variance of CSR features was mainly explained by donor identity and diagnosis but not by age, sex, sequencing batch, or treatment such as Ig supplementation (Fig. 4b and Supplementary Fig. 4a). We pooled reads from technical replicates to increase read numbers and rescue samples failing quality control. Various features significantly differed between CVID patients and controls including the fraction of single breaks (pct_direct_switch), the amount of sequence microhomology surrounding breaks (homology), the size of CSR clones (cluster_size_mean), and the fraction of clusters with Sγ2 or Sγ3 assigned isotypes (pct_clusters_SG2/SG3) (Fig. 4c). Interestingly, we found higher usage of Sγ3 and lower usage of Sα1 in CVID patients that did not elicit IgG titers after vaccination (Supplementary Fig. 4b). Similar to the reduced frequency of somatic hypermutation events in VDJ sequencing data from CVID patients[51], we found fewer somatic variants (somatic_variants) in our CVID patients than in the controls (Fig. 4c). In a PCA using the robust features, the controls separated well from CVID patients, as well as CVID-like disorders caused by specific IEI, such as deficiencies in *KMT2D* (Kabuki syndrome), *ICOS* (Inducible T cell costimulator), *AICDA* (encoding AID), *NFκB2* (nuclear factor kappa B subunit 2), and APDS2 (Activated PI3K-delta Syndrome Type 2) caused by loss-of-function mutations in the regulatory subunit of PI3K (*PIK3R1*) (Fig. 4d). The distinction between controls and immunodeficient samples, as seen in PC1, is for instance characterized by a high percentage of direct switch events without consecutive switches (pct_direct_switch), and a low percentage of Sγ3 clusters (frac_clusters_SG3; Supplementary Fig. 4c). As part of PC2, we observed increased direct switching to Sγ3 and increased Sα1 usage (pct_breaks_SM_SG, pct_clusters_SA1), low homology (homology), and a low percentage of Sγ4 clusters (pct_clusters_SG4).

Note that two patients (13 and 21), which were diagnosed as atypical CVID, have a potential failure in the plasmablast compartment, as they show low Igs in blood, but normal switched memory B cells. As expected, these samples clustered together with controls in the PC analysis, as did patients diagnosed with a subclass deficiency—except for an IgG4-deficient patient (Fig. 4d). Subclass deficient patients showed a significant reduction in the number of Sγ2 CSR clones (Supplementary Fig. 4d) but were otherwise comparable to controls. Next, we additionally obtained from the same laboratory 12 controls and 14 patients for a "CVID testing" dataset, including patients with mutations in *TNFRSF13B* (Tumor Necrosis Factor Receptor Superfamily Member 13B) encoding TACI (Transmembrane activator and CAML interactor), plus two additional biological replicates from control 09 and patient 32. From a different laboratory, we obtained the "DNA repair" set with 12 patients, including children, three of whom had a deficiency in *ATM*, three in *BRCA1*, one in *LIG4*, and one in *NIPBL*

(Nipped-B-like protein). In addition, 9 age-matched controls were obtained. Projection of the "CVID testing" (Fig. 4e) and "DNA repair" data (Fig. 4f) into the PCA of the "CVID training" samples validated a separation of control and patient samples of the two independent datasets.

Machine learning with ridge regression almost perfectly separated CVID and CVID-like patients from control samples of the "CVID testing" dataset, excluding replicate samples from training donors (Fig. 4g). Additional cross-validation with a 75:25 training:testing split resulted in a mean prediction accuracy of 0.92. Despite training on CVID data, DNA repair-related defects in the "DNA repair" samples were identified with an AUC score of 0.84, emphasizing a high degree of sensitivity and specificity of our predictor (Fig. 4g). Surprisingly, the IgA fraction measured by flow cytometry and SWIBRID correlated better in controls than in CVID samples, suggesting a potential alteration in B cell receptor (BCR) expression in CVID B cells (Supplementary Figs. 4e and 5).

Clustering of the "CVID training" and "CVID testing" cohorts resulted in three different groups (Fig. 4h and Supplementary Fig. 4f), with cluster 1 comprising mostly controls, atypical, and subclass deficient samples, and cluster 2 and 3 containing patients. Cluster 2, for instance, showed an increased frequency of untemplated inserts when compared to controls, as well as a very low amount of Sα CSR clones and high Sγ1, but otherwise less severely affected isotypes and diversity measures. Instead, cluster 3 showed a decrease for all Sγ isotype ratios and overall poor diversity measures, such as a low number of CSR clones and entropy (Fig. 4i). An interpretation of cluster 2 as immunocompromised versus cluster 3 as potentially DNA repair deficient was supported by shared features of cluster 3 and ATM deficiencies, such as decreased Sγ1, untemplated insertions and intraswitch deletions, as well as increased homology (Fig. 4i). This was further supported by inspecting PCA loadings and regression coefficients (Supplementary Fig. 4g, h) and by feature-class scores, which aggregate features from the initial categories (i) structural aberrations, (ii) sequence context, (iii) the breakpoint matrix, (iv) diversity measures and (v) isotype composition (Supplementary Fig. 4f). While PC analysis and clustering raises the impression that monogenic defects in e.g., *AICDA*, *TNFRSF13B*, or *ICOS* are not different from CVID patients, it is important to note that the CSR junction phenotypes are highly individualized (Fig. 4h). Although we did not establish a longitudinal cohort, samples from two individuals (09 and 32) collected over periods of up to four years clustered tightly in the heatmap and in PCA space (Fig. 4e), indicating a stable SWIBRID profile over time.

In conclusion, SWIBRID reliably identified CVID patients and monogenic defects, unravelling patient-specific CSR junction signatures, which allowed patient assignment to phenotypic clusters.

## Discussion

We developed SWIBRID to systematically interrogate immunocompetence and DNA repair by profiling the diversity of switched B cells and their CSR junctions. Our pipeline includes the identification of CSR clones and enables the quantification of a large catalogue of junction-related features, categorized into genomic aberrations, sequence context, breakpoint matrix, B cell diversity, and antibody isotypes.

In cell lines, we show that specific DNA repair defects exhibit distinctive CSR junction characteristics. Remarkably, SWIBRID-based phenotyping allowed the identification of specific DNA repair protein KOs in cell lines via machine learning. Compared to direct genotyping and classification of monogenic disorders, high-throughput CSR junction sequencing may be applied to unravel complex human defects caused by, e.g., mutations of unknown clinical significance, rare compound heterozygous traits, epigenetic defects, or synergistic effects of otherwise benign variants across genes. Technologies using WGS data, such as CHORD or HRDetect, identify HR deficiency in tumor tissues by analyzing mutational signatures across the

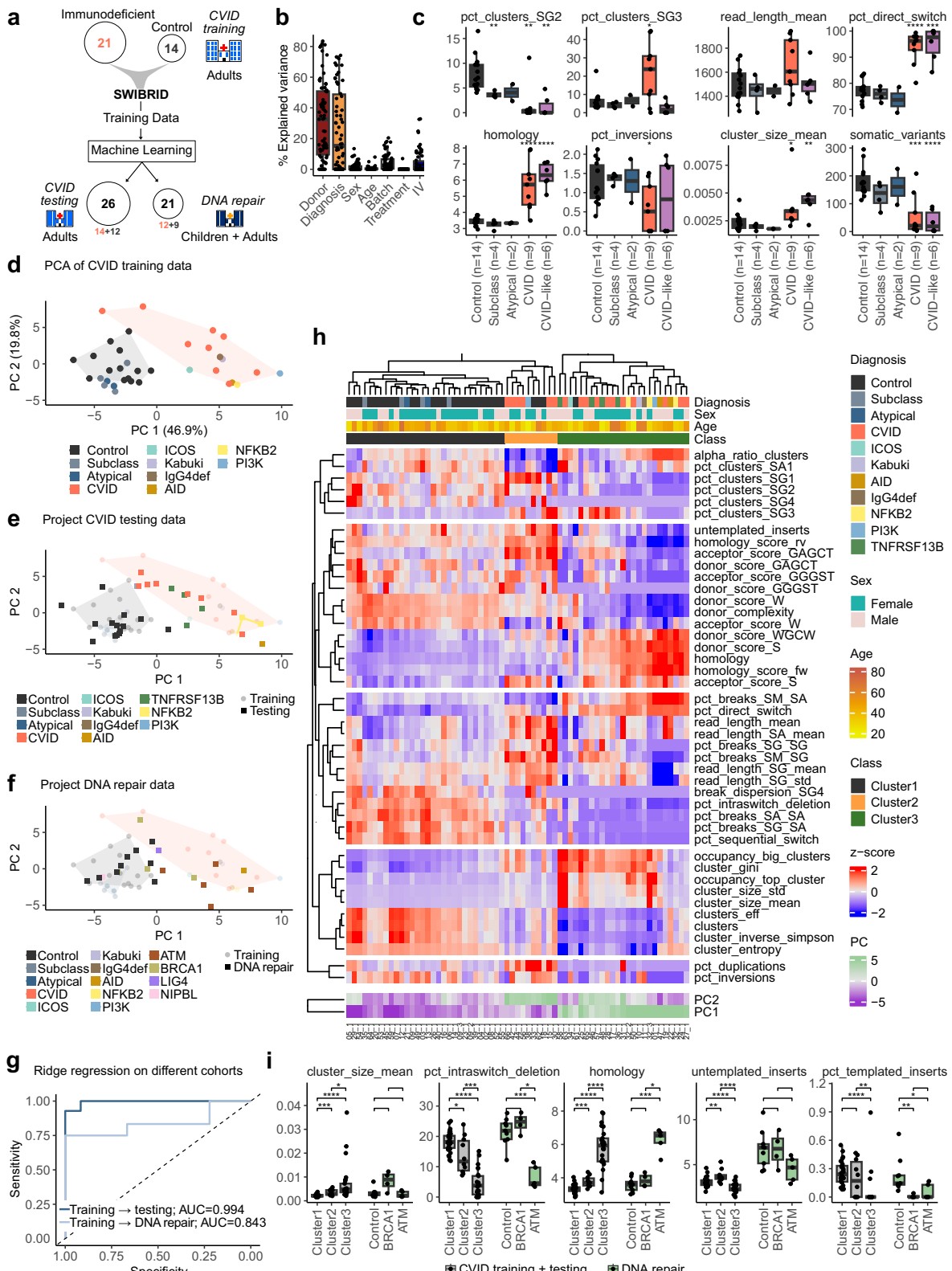

genome[22,23]. Similarly, SWIBRID identified features that changed in the context of HR deficiency in CH12 cells and human samples. However, as a blood-based assay, SWIBRID cannot be applied to tumor tissue to identify altered DNA damage repair due to tumor-intrinsic, somatic mutations.

In human samples, SWIBRID reliably identified individuals with CVID and DNA repair defects, even with scarce switched memory B

cells. Of note, in patients with very low amounts of B cells (AID-deficiency), SWIBRID was capable of identifying patient-specific signatures that are beyond the detection capabilities of currently used technologies, such as flow cytometry. Moreover, the granular signature of CSR junctions served to classify patients into so far uncharacterized groups. Given that CVID encompasses a heterogeneous group of disorders with complex phenotypes[52], CSR profiling has the potential to

**Fig. 4 | SWIBRID reliably identifies CVID samples and patient-specific CSR junction signatures. a** Overview of cohorts, consisting of a "CVID training" cohort (21 patients, 14 controls), "CVID testing" cohort (14 patients, 12 controls, and 4 additional samples of two donors from the training cohort), and "DNA repair" cohort (12 patients, 9 controls). **b** Explained variance for 70 features by donor identity, diagnosis, sex, age, batch, immunomodulatory treatment, or Ig administration (IV). **c** Box plots of selected features with data points representing 2–3 pooled technical replicates from $n = 35$ donors. CVID: patients with common variable immunodeficiency; CVID-like includes Kabuki, ICOS-, IgG4-, PI3K-, NFκB2-, AID-deficient patients. Atypical is used for CVID patients with suspected plasmablast defect. **d** PCA using 44 robust features, diagnosis indicated by color. **e** Data from the CVID testing cohort with 2–3 pooled technical replicates per donor were projected into the PCA of (**d**); longitudinal samples of 2 donors are included (connected by lines). **f** Projection of data from pooled replicates of the DNA repair cohort into the PCA of (**d**). **g** Receiver-operator characteristic (ROC) curve for the prediction of CVID and CVID-like (dark blue) or DNA repair (light blue) patients vs.

control donors using ridge regression in the CVID testing ($n = 26$) and DNA repair cohorts ($n = 21$), respectively. For the prediction, we used 44 features plus 29 samples for training. **h** Heatmap of scaled robust features for all donors of the CVID training and testing cohorts ($n = 65$ pooled samples from $n = 61$ donors). Top: Diagnoses, sex, age, and cluster class are indicated for each donor. Right: features are grouped by category according to Fig. 2a. Bottom: PC1 and PC2 coefficients from the PCAs of (**d**) and (**e**). Longitudinal samples from two donors are included: one healthy (09-0/1/2; collected 07/28/2020, 01/12/2016, 11/03/2020) and one CVID-like (32-0/1/2; collected 03/23/2023, 10/02/2023, 05/11/2020). **i** Box plot of representative features comparing cluster 1–3 as identified from the hierarchical clustering in (**h**) ($n = 63$) vs. controls, BRCA1- and ATM-deficient donors of the DNA repair cohort ($n = 21$). P-values in (**c** and **j**) from two-sided Wilcoxon test (****: $p < 0.0001$, ***: $p < 0.001$, **: $p < 0.01$, *: $p < 0.05$). Boxes in (**b**, **c**, and **j**) indicate 25th to 75th percentile; whiskers extend to largest/smallest value no further than 1.5× interquartile range, lines indicate median.

aid in more precise patient stratification. Interestingly, one CVID group exhibited signatures that are shared with ataxia-telangiectasia, which is caused by mutations in the ATM gene. This suggests that SWIBRID may be used to diagnose radiosensitivity and a higher cancer risk, which is a critical area of clinical importance in CVID[53].

In CVID, the primary defect lies in B cell maturation and terminal differentiation, with frequent reduction of switched memory B cells despite normal VDJ recombination. Recent studies suggest that rare, likely pathogenic variants in DNA repair genes such as *ATM*, *RAD50*, *NBS1*, *MSH2*, and *MLH1* may contribute to the pathogenesis of immunodeficiencies, and some of these factors (e.g., MLH1) are expected to affect the sequence context surrounding CSR junctions[54,55]. Furthermore, mutations in receptors expressed on B cells, such as TACI or BAFF-R, lead to distinct patterns of IgA and IgG deficiency[56,57]. Using SWIBRID, we capture both DNA repair-related alterations in CSR architecture and isotype composition, enabling a detailed analysis of B cell dysfunction beyond conventional immunophenotyping.

SWIBRID enables the identification and characterization of CSR defects, key features of CVID that may be missed by other clinical tests, such as TREC/KREC assays. For instance, despite their profound consequences for immune system dysfunction, diseases like ataxia-telangiectasia and Nijmegen breakage do not always result in poor TREC and KREC outcomes[58–60]. This may relate to the fact that such patients exhibit normal VDJ recombination of antigen receptors[58,61]. Likewise, other patients with antibody disorders or dysfunctional B and T cells may suffer from severe clinical phenotypes, but do not display abnormal results in TREC/KREC or VDJ repertoire[51]. When VDJ diversity is preserved, SWIBRID offers the advantages of focusing on switched B cells that only appear in the context of an immune response or a vaccination.

Our previous studies showed that antibodies can gain an extra domain from large templated insertions in the S region, but the precise molecular factors contributing to these events remain elusive[39,46,62]. Here, we observed in CH12 cells that *Rif1* KO strongly increased the frequency of templated inserts, while in human patients with BRCA1 mutation, we found a significant depletion of these inserts. However, this was not observed in CH12 *Brca1* KO. Of note, CH12 WT cells shared some but not all characteristics with human B cells, which could not be explained by their limitation to IgA switching. It likely reflects the different genomic architectures of S regions affecting the SWIBRID features in mouse versus human.

In conclusion, we here show that SWIBRID holds potential to advance our mechanistic understanding of DNA repair defects and may help to unravel the underlying molecular causes of 70-90% of immunodeficient patients that remain genetically undefined[63]. Beyond those with severe disease, identifying minor immune dysfunctions may help to better protect infection-susceptible individuals by guiding preventive measures such as vaccination.

## Methods

### Mouse and human samples

All mouse and human samples in this manuscript were obtained as part of studies approved by the ethical commissions mentioned in the sections below.

### CH12 B cells: cultivation, stimulation, and DNA extraction

The generation of CH12 cell lines[64] and knockout lines is described elsewhere: *Rif1* and *Lig4*[65], *Trp53bp1*[66], *Brca1*[44]. CH12 cells were cultured using B cell complete medium that contained: 10% FBS (PAN Biotech GMBH, #P30-3302), 1% non-essential aminoacids (Life Technologies, #11140035), 1% sodium pyruvate (Life Technologies, # 11360039), 1% GlutaMAX (Life Technologies, #35050038), 1% Penicillin-Streptomycin (Life Technologies, #15140163), 0.1% mercaptoethanol (Life Technologies, #31350010), 100 µg/ml Kanamycin (Serva Electrophoresis, #26897.01), and 0.002 % transferrin (Merck-Milipore, #616397).

For in vitro activation, 200,000 CH12 cells per milliliter were seeded in a 12-well plate (SARSTEDT, #833921). On day 0, 25 ng or 400 ng interleukin-4 (recombinant), 4300 ng of anti-mouse CD40 (BIOZOL, #BLD-102902), and 8.6 ng of mTGFβ1 (R&D Systems, #7666-MB-005/CF) were added to each well to induce B cell activation.

Cells were harvested on day 2 and gDNA isolation was performed using the ReliaPrep™ Blood gDNA Miniprep System (Promega, #A5082).

### Human samples and ethic votes

We used random 500 ml blood donations that were collected by the DRK-Blutspendedienst Nord-Ost GmbH to assess the diversity of CSR junction features in the population. The blood was tested to be free of HIV, HCV, and HBV. The use of these samples was approved by the Charité ethical commission in the ethics vote EA1/149/23. The corresponding consent requires us to use a de-identification strategy (randomize donor identity across different features) when making this data publicly available.

CVID patients and controls were obtained from the Uniklinikum of Freiburg and CVID patients fulfilled the diagnostic criteria for CVID (www.ESID.org). Genetic diagnosis for Kabuki syndrome, APDS2 (*PI3KR1*), *AICDA*, *TNFRSF13B*, and *ICOS* deficiencies were defined after whole exome or targeted sequencing. All experiments were performed under the ethical approval of local authorities 251/13 and 254/19 according to the Declaration of Helsinki. Some patients were undergoing treatments including: prednisone ($n = 3$), co-trixomazol ($n = 2$), cortisone ($n = 1$), steroids ($n = 1$), sirolimus ($n = 1$), and infliximab ($n = 1$). Ig supplementation was administered either subcutaneously ($n = 24$) or intravenously ($n = 8$).

Samples from patients with DNA repair defects were obtained from the Karolinska Institutet with the approval of the Institutional Review Board of Karolinska Institutet and the Ethics Committee of Tehran University of Medical Sciences (IR.TUMS.CHMC.REC.1398.030). The

samples included previously studied donors with deficiencies in *ATM*[67,68], *BRCA1*[38], as well as patients with mutations in *AICDA*, *LIG4*[36], and *NIPBL*[69] encoding genes and controls of similar age and sex[67,70]. Detailed information for all donors is provided in the Supplementary Data 3.

Written informed consent was obtained from all patients, healthy individuals, or their parents.

As stated in the results section, we pooled reads from up to 3 technical replicates of 86 samples to rescue samples failing quality control by increasing read numbers. Of note, beyond the samples described in the results section, we obtained samples from 5 additional patients (1 "CVID testing", 4 "DNA repair") that could not be rescued, likely due to extremely low B cell counts.

### Primary human blood cells: purification, sorting, and isolation of genomic DNA

PBMCs from 500 ml blood donations were isolated using Ficoll (Carl Roth GmbH, #0642.2) and centrifuging for 25 min at $500 \times g$ with an acceleration/break of 3/0 (Centrifuge Eppendorf 5910 R). Primary human B cells were isolated from PBMCs using CD19 Microbeads (Miltenyi Biotech, #130-050-301) following the manufacturers protocol. When B cell subpopulations were needed, class-switched memory B cells were FACS-sorted (CD27$^+$IgM$^-$IgG$^+$IgA$^+$). The following antibodies were used for staining: CD27-PE (dilution: 1:170, Miltenyi Biotec, #130-114-156, clone M-T271); IgD-PE-Cy7(dilution: 1:80, Miltenyi Biotec, #130-098-584); IgM-AF488(dilution: 1:1000, Life Technologies, #A21215); IgG-AF647(dilution: 1:500, Dianova, #109-606-170); IgA-AF647 (dilution: 1:500, Dianova, #109-606-01); IgA-APC-Vio770 (dilution: 1:170, Miltenyi Biotec, #130-113-473, clone IS11-8); CD19-Brilliant Violet 605 (dilution: 1.160, Becton Dickinson, #562653); CD20-PerCP-Vio700 (dilution: 1:170, Miltenyi Biotec, #130-113-377, clone LT20). Gating strategies used to sort memory human B cells can be found in Supplementary Fig. 6. PBMCs from CVID patients and controls were also isolated from fresh EDTA blood by Ficoll density gradient centrifugation following standard protocols. PBMCs were frozen in Iscove's Modified Dulbecco's Medium, 40% FBS, and 10% DMSO until further processing.

Frozen samples were gently thawed prior to DNA isolation. The thawing process involved the rapid melting of the cells using a water bath at 37 °C; and immediately after, the addition of cold B cell complete medium drop-wise into the vial to minimize temperature and osmotic shock. Thawed cells were resuspended in B cell complete medium at twice the original frozen volume. Cells were centrifuged and counted for further processing.

One million PBMCs were collected from the samples of C1 and CVID samples from the University of Freiburg for DNA isolation. DNA isolation was performed following the protocol of the ReliaPrep™ Blood gDNA Miniprep System (Promega, #A5082). DNA from one million PBMCs were eluted in 50 µl.

Genomic DNA (gDNA) was obtained from the DNA repair defects samples from the Karolinska Institutet following the "salting out procedure"[71].

### Mouse splenocytes and lymph nodes: isolation of cells and genomic DNA

Twelve female mice aged 9 to 10 weeks were immunized using HIV glycoprotein (gp) 140 on days 1, 21, and 42. On day 56, the mice were sacrificed and cells from spleen and lymph nodes were collected. This process was performed by Experimental Pharmacology & Oncology Berlin-Buch GmbH (Ther. E0023/23, study number: 18906). Between 900,000 and 2.5 million cells were collected per sample for DNA isolation. gDNA isolation from these cells was done following the protocol of the ReliaPrep™ Blood gDNA Miniprep System (Promega, #A5082).

### CSR junction PCR

The human CSR junction PCR included in one reaction 3 µl mM dNTPs each (Roboklon, #E0501-01), 5 µl LongAmpTaq Buffer, 1 µl LongAmpTaq polymerase (NEB, #M0323), nuclease-free water, primers and template until a total volume of 25 µl, 1 µl of 10 µM Sµ FW, Sα RV and Sγ RV primers[39], and optionally Sε RV, were added (Supplementary Table 1 and Supplementary Fig. 1b). The template used was either 200,000 PBMCs, in C1 and the CVID patients from the University of Freiburg; or 30 ng, from the DNA repair defects from Karolinska Institutet. The PCR protocol consisted in 25 cycles of 95 °C for 40 s, 60 °C for 30 s, 65 °C for 3 min; and the last elongation step at 65 °C for 10 min with a ramp speed of 4 °C/s using Biometra TAdvance (Analytik Jena). The use of a high-fidelity polymerase, along with optimized primer design, cycle number, and ramp speed, was carefully adjusted to minimize nonspecific binding, reduce the impact of secondary structures, and enable efficient amplification of highly repetitive, GC-rich switch joints.

The mouse CSR junction PCR included the same mixture of LongAmpTaq reagents and dNTPs as the human CSR junction PCR. However, mouse CSR junction PCR was performed separately for different combinations of forward and reversed primers (Supplementary Table 1 and Supplementary Fig. 1i). Thus, mSµ FW primer[72] was used in combination of mSα REV, mSγ2bc REV, mSγ3 REV, or mSγ1 REV in separate reactions, respectively. 100 ng of mouse gDNA from splenocytes, lymph nodes or CH12 was used in each PCR reaction. The PCR protocol consisted of 30 cycles of 95 °C for 40 s, 60 °C for 30 s, 65 °C for 3 min; and the last elongation step at 65 °C for 10 min with a ramp speed of 2 °C/s using Biometra TAdvance (Analytik Jena).

Amplicons were observed in an agarose gel and sporadically in the TapeStation using High Sensitivity D5000 ScreenTape (Agilent, #5067-5592) (Supplementary Figs. 1c–f, 7, and 8).

### BCR transcript analysis

BCR sequencing was performed as described in Turchaninova et al.[73] with several modifications. All primers used during this procedure are described in Supplementary Table 2. In brief, RNA was extracted from 50,000 or 100,000 memory B cells and used for cDNA synthesis with SMARTScribe RT (TaKaRa, #639538), using a mixture of constant domain primers at 1 µM final concentration and a SmartNNNext template-switch adapter. cDNA was purified with 1.8 volumes of AMPure XP beads (Beckman Coulter, #A63881) and eluted in 1 volume of milli-Q water (mQ). Purified cDNA was used as a template for 25 cycles of the first PCR with a mixture of nested constant domain primers and M1ss primer annealing to the template-switch adapter sequence at 0.2 µM. The PCR was purified with 1.2 volumes of ProNex beads (Promega, #NG2001) and eluted in 1 volume of mQ. Purified PCR was used as a template for 5 cycles of a second PCR with a mixture of Illumina adapters-bearing primers at 0.2 µM final concentration. The second PCR was purified with 1.2 volumes of ProNex beads and eluted in 1 volume of mQ. Purified PCR was used as a template for 5 cycles of indexing PCR with Illumina Nextera adapters. The products were pooled, purified with 1.2 volumes of ProNex beads, eluted in 1 volume of mQ, and analyzed by BioAnalyzer High Sensitivity dsDNA kit (Agilent Technologies, #5067-4626). The library was sequenced with MiSeq 600 cycles kit (Illumina, #MS-102-3003). Base-called data and demultiplexed data were analyzed with MiXCR[74] using the "generic-bcr-amplicon-umi" preset with the tag pattern "^N{22}(UMI:N{16})N{7}(R1:*)\^(R2:*)". The ratio of IGHA to IGHG was extracted from the allCHitsWithScore column in the final MiXCR output.

### Sequencing

CSR junction PCR products were purified using ProNex beads in a 1:1 ratio following manufacturer protocol and eluted in 16 µl, and 30% of amplicons were sequenced using MinION Nanopore (SQK-LSK109 and FLO-MIN106D) or PacBio Technologies (Genomics Facility, Max-Delbrück-Center, Berlin) following the manufacturers protocol.

## Bioinformatics pipeline

SWIBRID's bioinformatics pipeline is written in Python and Snakemake. It performs end-to-end processing of sequencing output (fastq files) to derived SWIBRID features. Code and documentation are available at bihealth.github.io/swibrid.

The pipeline consists of the following steps:

1) Demultiplexing

A custom demultiplexing script detects ONT barcodes as well as sequencing primers in the reads using BLAST with parameters optimized for short matches, and then distributes input reads into sample-specific output files according to sample barcodes, while collecting information of read length, quality, primers and barcodes detected at the read ends or internally (more than 100 nt from the ends) as well as other meta-data included in the input fastq headers into a separate csv file. The script features a "–split_reads" mode to detect clusters of multiple barcodes and primers in close proximity and split presumably concatenated reads into parts.

2) Alignment

Read alignment is performed using LAST (default; "last-train" followed by "lastal" and "last-split"), minimap2, STAR, or bwa. Alignments statistics in terms of mismatches, insertions, and deletions are obtained from "last-train" output or directly collected from the bam output of the other aligners. Optionally, reads are also aligned against telomer repeats using BLAST.

3) Processing and filtering

Reads are inspected for alignments to the switch (S) region or elsewhere (potential inserts). Isotypes are determined by overlap of the right-most alignment block with any of the annotated S regions. Reads are filtered out,

- that are too short
- that don't contain forward and reverse primers
- that contain internal primers
- that don't contain at least two separate alignments to the S region
- if mapped regions on the genome are much shorter than mapped parts of the reads
- if there is too much overlap between different alignments on the genome or on the read
- if too little of the read maps (less than 90%)
- if alignments to the S region are in the wrong order, or
- if an isotype cannot be determined.

(Templated) inserts are detected as alignment blocks to regions outside of the S region (or to the telomer repeats) flanked by alignments to the S region. Read orientation, coverage, fraction of read sequence mapping to the same genomic region, mapped S region segments, and filtered inserts are recorded. Additionally, the aligned read sequence (including gaps but ignoring insertions) is written out for construction of a (pseudo) multiple sequence alignment. Finally, 40 nt of sequence around each break between alignment blocks are taken from the read (20nt upstream and 20 nt downstream) and re-aligned against 40 nt of genomic sequence of donor and acceptor S region, using the PairwiseAligner routine from BioPython with penalties as used by minimap2 for MinION or pacBio reads, respectively. Untemplated nucleotides are counted as contiguous bases in the read near the break that don't align to either donor or acceptor region, homologous nucleotides are counted as contiguous bases near the break that align to both donor and acceptor region.

4) Construction of a (pseudo) multiple sequence alignment (MSA)

To facilitate efficient clustering of tens of thousands of reads of several kb length, we transform the reads aligned to selected S regions into a sparse coverage matrix of dimensions n_reads ×

n_bases. From here on, we use a reduced coordinate system of concatenated S regions (about n_bases = 30 kb in total) extended to the nearest multiple of 500 to allow for unambiguous binning. The matrix values m encode coverage and consensus nucleotide identity, such that m%10 encodes nucleotide identity (with A = 1, C = 2, G = 3, T = 4 and gaps encoded as missing or zero values), and m//10 encodes coverage of genomic regions by one read (typical values are 1, values of 2 or higher indicate tandem duplications, negative values an inversion). n_reads is typically constrained to at most 50,000 to limit computational resource usage and ensure that different samples are comparable.

5) Detection of gaps and junction breaks

Gaps in the MSA are detected and saved into arrays containing the read index, left and right position of the gap (in the reduced coordinates) and gap size.

6) Construction of a linkage by hierarchical clustering

Hierarchical clustering is performed using cosine metric and average linkage (UPGMA) on cleaned coverage patterns, i.e., the coverage values m//10 of the MSA with gaps smaller than a cutoff value (typically 75 nt) filled in to reduce noise from sequencing and mapping artefacts. Clustering can be done using the Python implementation of fastcluster on the dense version of the MSA, or by hierarchical agglomerative clustering with approximate average linkage on a nearest-neighbor graph, allowing sparse input and with improved time complexity and memory requirements (see Supplementary Note 1).

7) Defining and filtering clusters

Clusters are defined by cutting the dendrogram at a certain cutoff, which is typically fixed at 0.01 but can also be automatically determined by scanning cutoff values in a predetermined range and finding an inflection point in the curve that shows the cluster number as function of cutoff (either by distance from the straight line connecting the extremes or from the intersection of a double-exponential fit). Small and isolated (with large tree height of cluster root nodes) clusters are flagged such that remaining clusters contain at least 95% of the reads (see Supplementary Note 1).

8) Calling single-nucleotide variants

We implemented a routine to call single-nucleotide variants in the MSA by comparing observed nucleotide frequencies at a position with what is expected given the reference nucleotide and average mismatch frequencies estimated during the alignment step using a chisquare test. Variants are excluded if

- they are near gaps (more than 70% gap frequency in a 10 nt window)
- the minor allele frequency is lower than expected by a poisson model
- the strand bias is too high ($p$-value < 0.05 in a chisquare test and relative difference of read numbers in positive vs. negative direction >0.125)
- no cluster of at least 10 reads has an allele frequency larger than 0.4, or if the total allele frequency is below 0.4.

Variants are annotated using dbSNP. Next, filtered clusters with at least 10 reads are again grouped into "haplotypes" using weighted non-negative matrix factorization (wNMF package) of cluster-wise alternative allele counts with 2 components and cluster sizes at each position as weights. Co-segregating read clusters (i.e., putative haplotypes) are inferred from the component matrix, and variants are classified as likely homozygous germline if their allele frequency is >0.6 in both components, heterozygous germline if the allele frequency is >0.6 in one component and <0.6 in the other, and likely somatic otherwise. wNMF is run 100 times and a consistency score is derived by counting how often clusters are sorted into the same component.

9) Calling structural rearrangements

Structural variants (rearrangements) are called from the MSA by finding regions with coverage >1 (duplications) or <0 (inversions) without gaps bigger than 25 nt. Arrays of read indices, left and right positions, and sizes are returned for duplications and inversions separately.

10) Analysis of clustering results

Aggregate statistics obtained from the clustering results:

- cluster size (number of reads)
- cluster isotype (most frequent)
- average read fraction mapped per cluster
- average read fraction mapped to same region multiple times
- average read fraction ignored because it maps outside the defined switch regions
- consensus sequence
- length of consensus
- GC content of consensus
- spread of breakpoints (number of bases with intermediate coverage)
- size of inversions
- size of duplications
- frequency of insert-containing reads in cluster
- fractional overlap of inserts in cluster
- fractional overlap of insert-associated breakpoints
- average insert length
- average length of gaps around inserts
- average isotype associated to inserts
- number of homologous nucleotides around insert-associated breakpoints
- number of homologous nucleotides around switch region breakpoints
- number of untemplated nucleotides around insert-associated breakpoints
- number of untemplated nucleotides around switch region breakpoints
- length of different S regions covered in cluster
- cluster size adjusted for PCR length and GC bias

11) Downsampling of clustering

For more robust estimation of diversity measures, the clustering is downsampled 10 times on a subsample of typically 1000 reads, using fastcluster with cosine metric and average linkage and the same cutoff as used in the cluster definition step. The following diversity measured are calculated:

- cluster_size_mean: average fraction of reads in clusters
- cluster_size_std: standard deviation of cluster size
- clusters: number of clusters after filtering
- clusters_eff: from the entropy of the cluster size distribution (pre-filtering)
- cluster_gini: Gini coefficient (post-filtering)
- cluster_entropy: entropy (post-filtering)
- cluster_inverse_simpson: inverse Simpson coefficient (post-filtering)
- occupancy_top_clone: fraction of reads in the biggest cluster
- occupancy_big_clones: fraction of reads in clusters with more than 1% of reads

12) Breakpoint statistics

A 2D breakpoint histogram matrix is generated from the detected gaps ≥75 nt counting the number of reads with donor and acceptor position in 50 nt bins along the reduced coordinate system. Reads are weighted according to cluster size such that all clusters contribute equally to the histogram unless specified otherwise. This will produce various aggregate statistics on breakpoints:

- breaks_normalized: number of breaks divided by number of clusters (or reads)

- pct_duplications: percentage of breaks leading to duplication events
- pct_inversions: percentage of breaks leading to inversion events
- pct_direct_switch: percentage of breaks from $S_\mu$ to another S region
- pct_sequential_switch: percentage of breaks between two different S regions, but not SM
- pct_intraswitch_deletion: percentage of intra-switch deletions within the same S region
- pct_intraswitch_inversions/duplications: percentage breaks with inversions or duplications within the same S regions
- pct_breaks_X_X: percentage of breaks connecting indicated S regions
- break_dispersion_XX: dispersion (standard deviation) of breakpoint positions in an S region
- homology_score_fw: homology in bins around breakpoint positions (same orientation)
- homology_score_rv: homology in bins around breakpoint positions (opposite orientation)
- homology_score_fw/rv_XX: homologies for breaks connecting indicated S regions
- donor/acceptor_score(_XX): motif scores for donor and acceptor breakpoints for different motifs (subdivided by S region)
- donor/acceptor_complexity(_XX): sequence complexity for donor and acceptor breakpoints (subdivided by S region).

Homology scores are calculated from 2D binned breakpoint frequencies weighted by homology of bins, motif scores are calculated from 1D binned breakpoint frequencies weighted by motif occurrences.

13) Read plots

Using an MSA and clustering results, this routine displays individual reads mapping over the S regions. Reads will be ordered as dictated by the linkage, or simply by isotype and cluster value. Reads can be colored by different variables (isotype, cluster, haplotype, coverage, strand, orientation, or other columns present in an additional info file). Sidebars can display additional variables per read or cluster. Variant positions or breakpoint realignment statistics can also be indicated.

14) Summary statistics

This routine produces a summary of features derived from a sample. Statistics produced by process_alignments, find_clusters, and downsample_clustering are collected. Cluster-specific values from clustering analysis and breakpoint statistics are averaged over a sample. Number of reads/clusters per isotype, and statistics on variants (germline vs. somatic, transitions vs. transversions, etc.) are collected.

### Generation of simulated switch junction reads

Simulated reads were generated using the "swibrid test" mode of the SWIBRID pipeline. In short, given a bed file with alignment coordinates of SWIBRID clones from a pool of 31 control donors on a truncated genome containing only 1MB of chr14, the script will randomly choose n entries of the bed file, obtain the concatenated genomic sequence for each entry, create an average of *k* copies according to a specified distribution (default: poisson), and add mutations, insertions, and deletions according to parameters estimated by LAST from a non-PCR control plasmid sample.

### Vincendeau et al. re-analysis

Sequencing data of CSR junctions generated by Vincendeau et al.[30] were downloaded from SRA (accession number PRJNA831666) and processed with standard SWIBRID settings for mouse, after adjusting coordinates of the relevant switch region for different primer locations.

## HTGTS data re-analysis

Sequencing data of CSR junctions generated with by HTGTS[27] were downloaded from SRA (accession numbers SRR2104731-44 for spleen, SRR6293456-79 for CH12). Overlapping paired-end reads were merged using bbmerge and then subjected to SWIBRID analysis, using bwa-mem2 (v2.2.1) for alignment. We adapted parameters of the processing and filtering steps to extract reads with SA, SG1, or SE isotype and at least two S region matches, allowing for up to 10 nt overlap between alignments on the read. We then constructed the MSA over suitably adapted switch regions, did not remove gaps, but only re-aligned reads across junctions that were at least 100 nt apart.

## Downstream analysis

Pipeline configuration files and analysis code for this manuscript are available at github.com/bihealth/swibrid_paper. Pipeline results for all human and mouse samples were connected with relevant meta-data. Samples with less than 500 final reads or otherwise failing visual post-pipeline quality check (QC) were removed.

For variance partitioning, we used the R package variancePartition[75] (v1.32.5) and a design "~ (1 | donor) + (1 | age) + (1 | sex) + (1 | sequencing) + (1 | batch)" for Fig. 2, or "~ (1 | donor) + (1 | diagnosis) + (1 | sex) + age + (1 | batch) + (1 | Treatment) + (1 | IV)" for Fig. 4.

PCAs were calculated on the selected robust features after z-score normalization using the prcomp R function, adapting isotype names between human and mouse, and removing irrelevant features for data from Vincendeau et al. (related to Sα or Sγ2/Sγ3/Sγ4) or from CH12 samples (related to Sγ, sequential breaks, isotype composition). Test or validation data were independently normalized and projected into the PCA.

Heatmaps were created with the ComplexHeatmap R package (v2.18.0)

Ridge regression was performed with the glmnet R package (v4.1-8), using the cv.glmnet function to estimate an optimal value of the regularizer. Training, test and validation data sets were independently normalized using z-scores. For the CH12 dataset, we used 33 features plus 49 samples for training and 36 samples for testing. For the CVID dataset, we used 44 features plus 29 samples for training, 26 samples for testing, and 21 samples for validation.

## Reporting summary

Further information on research design is available in the Nature Portfolio Reporting Summary linked to this article.

## Data availability

Raw sequencing data from mouse samples are available at SRA via accession number PRJNA1190672. Raw data for human samples cannot be shared publicly in accordance with patient privacy legislation. SWIBRID output for these samples, anonymized as required by the respective Ethics statements, is available at github.com/bihealth/swibrid_paper (https://doi.org/10.5281/zenodo.17415060). Source data are provided with this paper.

## Code availability

SWIBRID code and documentation are available at github.com/bihealth/swibrid (https://doi.org/10.5281/zenodo.17415052) and bihealth.github.io/swibrid. Scripts used to analyze data in this manuscript are available at github.com/bihealth/swibrid_paper (https://doi.org/10.5281/zenodo.17415060).

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

## Acknowledgements

We are particularly thankful to Cathrin Gerhard for her invaluable technical assistance throughout this study. We are grateful to all members of AG de la Rosa, in particular Lisa Spatt, Casper Silvis, Ata UW. Ahmad, Emre Mert İpekoğlu, and Dennis J. Doorduijn for their collaborative spirit and help. We thank members of AG Beule for productive discussions. We would like to thank Caroline Anna Peuker for collecting samples. We are grateful for the technical support provided by the Genomics Facility of the Max-Delbrück-Center, Berlin. Finally, we extend our gratitude to Klaus Rajewsky for his invaluable feedback and insightful discussions. This work was funded by German Research Foundation (394523286, to K.D.L.R.); The Helmholtz association; the Berlin Institute of Health at Charité, the Stiftung Charité, and the European Research Council (ERC) under the European Union's Horizon 2020 Research and Innovation programme (948464), all awarded to K.D.L.R.

## Author contributions

C.V.G. established the experimental system and performed the experiments, analyzed and interpreted the data, and wrote the manuscript. B.O. developed the SWIBRID code, analyzed, interpreted, and visualized the data, and wrote the manuscript. M.L. performed VDJ sequencing and analyzed the data. C.R. performed in vitro assays. B.K., A.B., M.D.V., S.M., D.S., Q.P.H., H.A., and K.W. provided cell lines or patient samples with clinical data and contributed to discussions. D.B. provided access to the BIH compute cluster and contributed to discussions. K.D.L.R. acquired the funding, analyzed, and interpreted the data, wrote the original draft, supervised and conceptualized the project. All authors read and approved the manuscript.

## Funding

## Competing interests

The Max Delbrück Center for Molecular Medicine (MDC) and the Berlin Institute of Health at Charité (BIH@Charité) have filed a patent application in connection with this work on which C.V.G., B.O., and K.D.L.R. are inventors (PCT/EP2024/057150). The remaining authors declare no competing interests.

## Additional information

¹Max Delbrück Center for Molecular Medicine in the Helmholtz Association, Berlin, Germany. ²Charité – Universitätsmedizin Berlin, Corporate Member of Freie Universität Berlin and Humboldt-Universität zu Berlin, Berlin, Germany. ³Core Unit Bioinformatics, Berlin Institute of Health at Charité – Universitätsmedizin Berlin, Berlin, Germany. ⁴Department of Rheumatology and Clinical Immunology, Medical Center - University of Freiburg, Faculty of Medicine, University of Freiburg, Freiburg, Germany. ⁵Center for Chronic Immunodeficiency (CCI), Medical Center - University of Freiburg, Faculty of Medicine, University of Freiburg, Freiburg, Germany. ⁶Division of Immunology, Department of Medical Biochemistry and Biophysics, Karolinska Institutet, Stockholm, Sweden. ⁷Research Center for Immunodeficiencies, Pediatrics Center of Excellence, Children's Medical Center, Tehran University of Medical Sciences, Tehran, Iran. ⁸German Cancer Consortium (DKTK), partner site Berlin, a partnership between German Cancer Research Center (DKFZ) and Charité – Universitätsmedizin Berlin, Berlin, Germany. ⁹Experimental and Clinical Research Center (ECRC), a junction cooperation between Charité and MDC, Berlin, Germany. ¹⁰Department of Gynecology with Breast Center, Hereditary Breast and Ovarian Cancer Center, Charité – Universitätsmedizin Berlin, Berlin, Germany. ¹¹Berlin Institute of Health at Charité – Universitätsmedizin Berlin, Berlin, Germany. ¹²Centre for Individualised Infection Medicine (CiiM), Hannover Medical School (MHH), Hannover, Germany. ¹³Department of Personalized Immunotherapy, Centre for Individualized Infection Medicine (CiiM), Helmholtz Centre for Infection Research (HZI), Hannover, Germany. ¹⁴These authors contributed equally: Clara Vázquez García, Benedikt Obermayer. ✉e-mail: Kathrin.delaRosa@helmholtz-hzi.de

