## [Transparent Peer Review file · Nature Communications]

Recombination junctions from antibody isotype switching classify immune and DNA repair dysfunction

Corresponding Author: Professor Kathrin de la Rosa

Version 0:

Reviewer comments:

Reviewer #1

(Remarks to the Author)

The authors PCR immunoglobulin class switch recombination (Ig CSR) regions from pools of human peripheral blood mononuclear cells (PBMC). They then do long-read sequencing with the goal of analyzing the Ig switch junctions for features. They have a computational algorithm that combines junctional features and other parameters to allow them to plot their Principal Component (PCA) 1 versus PCA2. I have the following comments and suggestions.

1. I commend the authors analytical skills, but the writing places this paper beyond the analytical ability of all but a fraction of the bioinformatics community and beyond any molecular biologists. I would say the same thing about similarly informatically-intensive papers by others in other journals, but I do not review all such papers. I wish to be helpful to the authors because I respect all the effort that they have put into this fine work.

2. The PBMC pools consisted of up to 200,000 cells per analysis. Of such a pool, they estimate it contains 10,000 B cells. How many cycles of PCR do they do prior to the sequence analysis step? I am worried about complications during the PCR steps at regions of repetitive DNA. The switch regions are distinctive for their repetitive sequence. Please show me a PacBio read after a minimal number of cycles (e.g., 8 cycles) versus whatever number they normally do (I could not find a statement of the number of PCR cycles in the paper). Also, if they authors amplify a much smaller number of cells, how does this compare to a larger number of cells from the same patient.

3. If the authors take the PBMC from one patient and divide in half and the PCR, and sequence, please show how similar or different the results are. For example, can they find the top 10 largest and top 10 smallest deletions in each of the two halves of this analysis?

4. Beyond the one or a few actual sequence junction examples shown, please provide 20 actual human patient junctions and the sequence features of each of these (from the pooled PBMC samples). I request this from all bioinformatic DNA sequence studies.

5. Because Ig CSR sequences are very repetitive, assessment of microhomology and perhaps even some other features may be difficult.

6. The use of the term 'scar' is inappropriate at all places in this manuscript. These are simply genomic alterations. Near the very end of PMID 36587186, that author explains the inception of the phrase 'information scar', which was not to be confused with any physical 'scar' in the DNA duplex structure. Since that inception, the dramatic use of the term 'scar' even for simple mutations has become disturbingly frequent in the literature, even though this is inappropriate, especially for non-coding regions such as the Ig CSR regions which do not encode information. I provide this comment for all papers that I review that inappropriately use the phrase 'genomic scar'.

7. In the Supplementary file, lines 25-26, the authors write: 'Given a high rate of sequencing errors, only the coverage of a position (not nucleotide identity) was considered, and gaps smaller than 75nt were ignored.' Given the repetitive nature of Ig CSR sequences, could the authors explain this to me a bit more in their responses?

8. For COVID, it is my understanding that this has diverse causes that do not affect the actual DNA events of VDJ recombination or class switch recombination. Perhaps in the Supplement, they authors could comment on how or why the analysis gives a readout beyond simply the number of B cells that have an antigen receptor.

Again, I commend the authors for their impressive amount of work.

(Remarks on code availability)

Reviewer #2

(Remarks to the Author)

The manuscript by Vázquez García, Obermayer et al., presents a novel method named SWIBRID (SWItch joint Breakpoint Repertoire IDentification) designed to characterize DNA double-strand break (DSB) repair outcomes during antibody class switch recombination (CSR). Leveraging decades of foundational research into B-cell mediated DNA repair, this method uniquely profiles genomic scars resulting from CSR, allowing for assessment of immune diversity and DNA repair functionality from peripheral blood samples. Employing long-read sequencing and sophisticated bioinformatics pipelines, SWIBRID identifies distinct repair signatures in various DNA repair mutants and in patients with immunodeficiencies such as Common Variable Immunodeficiency (CVID) and cancer-associated DNA repair deficiencies. The authors demonstrate discriminatory power in distinguishing repair-deficient genotypes and patient cohorts, underscoring potential clinical utility in personalized diagnostics and early detection of pathogenic defects. Some issues still requiring clarifications include:

- 1- The methodology shows promise in distinguishing specific DNA repair deficiencies and CVID. Have the authors considered potential confounding factors such as age-related changes in B-cell repertoire or treatment-related effects on CSR in clinical samples?
- 2- It remains unclear whether the CSR scar patterns detected by SWIBRID are stable over time in individual patients or influenced by transient immune activation events or infections. Are there any longitudinal patient data to draw such conclusions from in the analysed sample cohort?
- 3- While the method categorizes patients effectively, it would be highly beneficial to correlate specific CSR signatures with clinical outcomes or disease severity, if possible. Such correlation would substantiate clinical relevance and utility.
- 4- The manuscript introduces several CSR-related genomic scar features. However, the biological mechanisms underlying some of these newly identified features (e.g., long templated insertions, certain sequence context motifs) require further experimental validation to clarify their roles in DNA repair pathway preference or dysfunction, or at least additional discussion to their origins and potential mechanisms.
- 5- The manuscript employs machine learning (ridge regression) to classify genotypes. Detailed clarification on model robustness, cross-validation methods, and generalizability to external datasets would be ideal.

Minor Concerns:

1. Line 141–147: clarify units—reads or cells?
2. Extended Data Fig. 1f,g legend missing information to be understandable.
3. Figure quality: many diagrams use colour-only encoding (e.g., PCA). Or have overlapping color legends with such as fig 2b.
4. Methods: provide accession for training datasets (mouse Vincendeau et al.)
5. Some sections, particularly those presenting comparisons and statistical significances, could benefit from clearer methodological descriptions, including exact statistical tests used and justification for the chosen methods.
6. The authors acknowledge technical noise from sequencing errors. More explicit discussion and mitigation strategies for these technical limitations, including error rates of Nanopore versus PacBio technologies, are essential for readers to interpret data quality accurately.
7. Discussion on how SWIBRID might compare to existing clinical tests (e.g., TREC/KREC assays) could provide valuable context for clinical integration.

In conclusion, the manuscript significantly advances the field by introducing a powerful new method for profiling DNA repair competency and immunological health through CSR scars. Addressing the highlighted concerns will enhance the robustness, clarity, and broader impact of this promising study.

(Remarks on code availability)

Reviewer #3

(Remarks to the Author)

The authors of Garcia and Obermayer et al describe SWIBRID to identify the repertoire of switch joint breakpoints that uses PCR and long read sequencing from blood samples to detect DSB repair patterns driven by IGH class switch recombination (CSR) in the B cell subgroup. In surveying patient samples, the authors report immunodeficiencies, DNA repair defects, and patient-specific scarring signatures with high accuracy. The work looks to be well developed with many identified features that may be useful for sub-group distinction across human and mouse samples but this reviewer was struggling to figure out which new features were the most frequent and most relevant to the CSR repair outcome repertoire. The manuscript could be improved further by addressing the following comments below:

Major Comments

Most of the 68 features in 5 categories are presented as fractional data, presumably to normalize for absolute variance measures. It would be very helpful to have all of the data outlined in the figures mirrored in absolute frequencies in supplemental tables; this would be analogous to showing the films of blots. It is important for readership to evaluate which features are weighted by high frequency, the rigor of the sequencing data, and to confirm any statistically significant shifts in proportional data are not influenced by low frequencies which may make some highlighted features too noisy for evaluation.

Adding to the first comment, too many features will naturally yield some measurable difference that may not be related to biology unless the magnitude of difference withstands the increased number of multiple comparisons. As the central point of the manuscript is to highlight the numerous features, then proper statistical rigor needs to account for false discovery.

Fig2a is somewhat helpful for revealing the categories but a detailed description of the full 68 features should be outlined in a supplemental figure/table so readers can fully evaluate the data presented. The feature names are bulky. Some terms were not readily understood by this reviewer (i.e. not clear from only looking at the figures as a litmus test for comprehension) and will certainly alienate most readers. Please reconsider renaming for efficiency and clarity that could pair with detailed description of the filters/ranges used to define each feature. Fig2h and related would certainly benefit.

Furthermore, for each experiment using the feature/category breakdown, a list of ranked features impacting the experiment should be provided. If none are impacting, then indicate none (presume this is the case for Spleen Lig4 in EDFig3b? If not need to indicate in figure legend why no Lig4 data for spleen). In this regard, it is difficult to follow the logic as to why 32 features were used for one set of knockouts and 39 features in another, what those features were and why others were excluded; it would appear that the remaining 20 or so have little value if they are not used frequently.

In terms of PCA, you need to indicate which components distinguish each of your comparisons and which features overlap. This is otherwise generically saying there are differences and similarities or they are separated from controls, which have no added value to the field given the prior work on the genes implicated or to provide causal sources for distinction in CVID/CVID-like samples; this latter part is partially done but needs to be applied to earlier parts.

Related to point #5 is that a laundry list of features are indicated but needs to be distilled down to what those robust affecting features mean in aggregate.

Methods section does not describe machine learning and more detail is needed beyond multinomial ridge regression for each experiment they were used. For independent reproduction of findings, the specific features used for each prediction experimental sets should be listed and any additional values, range for confidence, number of technical versus biological replications, etc. in the model should be reported. If this software is not cited in MS, please make sure this is included.

Minor Comments

This manuscript suffers from poorly framed figures with text and symbols that are either too crammed together, too small, or using inappropriate axis labeling. For the latter, there are no such fractional n_{homology} measures and listing 0.5 increments only serves to cause confusion to the lay reader. Labeling format (e.g. n_{homology} , Fraction IGA VDJ) may be understandable as a programming label for the authors but is not for lay audience. This reviewer urges the authors to carefully consider the context of the data shown and reformat the figure panels to make it more conceptually available to a broader audience.

Not clear what type of V(D)J sequencing was occurring in the study as it is noted in the manuscript. Fig 1F labeling is also not clear. Was any variable region analysis performed and how was that integrated into SWBRID comparison? Line 152 is related to the same panel but in comparison to IGHA vs IGHG. There seems to be a disconnect in the text and figure.

There is a disconnect in feature numbers for iii on line 163 versus Fig 2a. Please clarify.

Lines 171-172. Presuming only data from 1 of the 10 batches are relevant given that was the only batch using long read sequencing, please make sure to break down that information when comparing this to C2. If using both sequencing formats, then indicate as such for both cohorts. Only 6 of the 68 features are referenced; why those 6? Are they representing the majority of the data? If not, then why only show a minority? Brief explanation in the main text or more detailed explanation in the Supp. Note may be useful.

PCAs need to have indicated in legend how many robust features were ultimately used since this seems to change across experiments.

For Fig2h and EDFig3a graphs you are better off showing only those with FDR-adjusted significance as showing everything despite some with larger symbols is uninterpretable and providing too much unnecessary information.

Lines 260-261: not clear whether this new observation of GC content on the acceptor side of the old data that is not shown is conserved in your experiential data on Lig4 deficiency in the prior paragraph.

Lines 287-288: Can the authors speculate in the discussion as to what the correlation means?

V(D)J is more accurate than VDJ and symbolizes that some antigen receptor loci (i.e. lambda, kappa, alpha, gamma) only

use V-J recombination.

(Remarks on code availability)

Version 1:

Reviewer comments:

Reviewer #1

(Remarks to the Author)

Reviewer 1.

My goal is to continue to support publication. I respect the authors responses to my questions. Importantly, the authors are applying long-read sequencing to a repetitive, GC-rich region, as they acknowledge. So their experience with difficulties is part of the VALUE of this paper. I strongly suggest (almost insist) that the authors include in their Supplement the figure that they provided in their response to me (Fig. X1), along with the text (and legend) they wrote associated with X1: "the number of PCR cycles was a critical parameter requiring careful optimization. On the one hand, we needed sufficient amplification to enrich CSR junctions and generate enough material for the minION library preparation; on the other hand, we aimed to keep the cycle number as low as possible to maintain diversity. When PCR is performed in regions of repetitive DNA, recombination can create artificial diversity, which in a monoclonal cell line would appear as heterogeneous outcomes. We do observe some diversity after 25 PCR cycles (Figure X1A-B). However, our read plot (Figure X1A) and sequence alignment of 8 random reads (Figure X1C) still show considerable homology in a monoclonal sample with two switched alleles, including i) highly similar read alignments to the switch regions (Figure X1A) and ii) substantial nucleotide homology in the alignments (Figure X1C), enabling precise breakpoint identification. As PCR-induced artifacts increase with cycle number, we conclude that 25 cycles under our optimized conditions introduce only minimal artificial diversity."

I think that the diversity in the black histogram peaks that is clear in Figure X1 (panel A, left upper edge) seems to indicate to my naive mind that the amplification of this monoclonal cell line gives heterogeneous switch junctions at each of the two alleles, especially the upper one. Or do these authors have a different interpretation of this? Their comments on this are valuable regardless of their response. They should rest assured that I am not going to use their response against them. But readers, along with me, deserve clarity on this point, and I think these excellent authors also wish to be clear as well.

(Remarks on code availability)

Reviewer #2

(Remarks to the Author)

The authors have satisfactorily addressed my concerns.

(Remarks on code availability)

Reviewer #3

(Remarks to the Author)

The authors made substantial improvements throughout the manuscript. This reviewer has no additional constructive comments to add.

(Remarks on code availability)

Version 2:

Reviewer comments:

Reviewer #1

(Remarks to the Author)

The authors seem to have responded to my comments, though in a somewhat minimal manner. I am ok with it.

(Remarks on code availability)

We would like to thank the reviewer for the overall positive evaluation of our work and for the valuable feedback, which has helped us improve the manuscript. While we acknowledge that a substantial portion of the manuscript is focused on bioinformatics, we deliberately chose *Nature Communications* over a dedicated bioinformatics journal due to the broad relevance of our method to the fields of DNA repair and immunology, as well as its direct applicability to patient samples. We have carefully addressed all points raised, as outlined in our point-to-point response.

Reviewer #1:

1. I commend the authors analytical skills, but the writing places this paper beyond the analytical ability of all but a fraction of the bioinformatics community and beyond any molecular biologists. I would say the same thing about similarly informatically-intensive papers by others in other journals, but I do not review all such papers. I wish to be helpful to the authors because I respect all the effort that they have put into this fine work.

While we have tried to present our ideas as clearly as possible, we regret if our writing appears incomprehensible at places. We fully share the goal of making our analytical approach accessible to non-specialists, but face the challenge that high-throughput methods, such as ours, need rather complex bioinformatics pipelines to address systematic biases and derive robust output. Different from other fields in genomics, where highly non-trivial methods for variant calling, gene expression quantification, or differential expression testing have become standard, here we could not rely on established methods and needed to implement a completely de-novo approach.

In response to the comments from all three reviewers, we have revised the manuscript to better explain our bioinformatics approach, as described below. For instance, we added additional explanations on the features terminology (see TableS1_features and materials and methods line 629-694). In addition, we better explained experimental setups and pipelines within the results section (e.g. lines 136-139, 153-157, 178-182) and improved figure legends (e.g. legend of figures 2 and 3, lines 821--824, 831-833, 840).

2. The PBMC pools consisted of up to 200,000 cells per analysis. Of such a pool, they estimate it contains 10,000 B cells. How many cycles of PCR do they do prior to the sequence analysis step? I am worried about complications during the PCR steps at regions of repetitive DNA. The switch regions are distinctive for their repetitive sequence. Please show me a PacBio read after a minimal number of cycles (e.g., 8 cycles) versus whatever number they normally do (I could not find a statement of the number of PCR cycles in the paper). Also, if they authors amplify a much smaller number of cells, how does this compare to a larger number of cells from the same patient.

The reviewer raises an excellent point, and we agree that PCR amplification of repetitive, GC-rich regions is inherently biased due to their lower amplification efficiency. This bias can only be partially mitigated, and we carefully considered it when establishing our initial protocol. To reduce the impact of secondary structures, we used a high-fidelity polymerase, optimized primer design, and adjusted the ramp speed to minimize nonspecific binding (which is now stated in the methods

section: “The use of a high-fidelity polymerase, along with optimized primer design, cycle number, and ramp speed, was carefully adjusted to minimize nonspecific binding, reduce the impact of secondary structures, and enable efficient amplification of highly repetitive, GC-rich switch junctions.”). As the reviewer correctly notes, the number of PCR cycles was a critical parameter requiring careful optimization. On the one hand, we needed sufficient amplification to enrich CSR junctions and generate enough material for the minION library preparation; on the other hand, we aimed to keep the cycle number as low as possible to maintain diversity.

When PCR is performed in regions of repetitive DNA, recombination can create artificial diversity, which in a monoclonal cell line would appear as heterogeneous outcomes. We do observe some diversity after 25 PCR cycles (Figure X1A-B). However, our read plot (Figure X1A) and sequence alignment of 8 random reads (Figure X1C) still show considerable homology in a monoclonal sample with two switched alleles, including i) highly similar read alignments to the switch regions (Figure X1A) and ii) substantial nucleotide homology in the alignments (Figure X1C), enabling precise breakpoint identification. As PCR-induced artifacts increase with cycle number, we conclude that 25 cycles under our optimized conditions introduce only minimal artificial diversity.

Figure X1. Clustering and sequence alignment of a monoclonal sample amplified with 25 cycles. A. Readplot of 2,500 reads after SWIBRID analysis of a monoclonal B cell line obtained from EBV immortalization containing two switched alleles. **B.** Plot depicting clones from the samples in A. Every circle represents a clone with a unique CSR junction. The size of the circle corresponds to the number of reads. Colours are matched with A. **C.** Manual analysis of 8 reads from the monoclonal sample. Break shows alignment in the big deletion found in switch μ region. Coordinates belong to hg38 genome assembly. A and B taken from Fig 1.

A

B

Clone 1:

```
SM - chr14: 105860018-105860108
AAGGCTAG-/GCTGAGCTGAGCTGGGCTGAGCAAGGCTAGGCTGAGCTGAGCTGAGCTGGG/CTGCGCTGAGCTGGGCTGGGCT
AAGGCTAG-/GCTGAGTTGGGCTAGCTGACTGGGCTAGGCTGAGCTGAGCTTGGTTGGG/TTGACCTGGGCTGAGCTGAGCT
AAGGCTAG/A/G---GCTGAGCTGAGCTGGGCTGAGCTGAGCT-TGCT-AG-----/TTGACCTGGGCTGAGCTGAGCT
AAGGCTAG-/GCTGAGCTGAGCTGGGCTGAGGAGGCTAGGCTAGGCTGAGCTGAG-----/-----CTGGGCTGAGCTGAGTT
TGGGCTGA-/GCTGAGCTGGGCTGAGCTGGACTGGGCTAGGCTGAGCTGAGCTTGGTTGGG/TTGACCTGGGCTGAGCTGAGCT

SA1 - chr14: 105710403-105710483
```

Figure X2. Cluster tracing in samples of one donor amplified using 15, 20 and 25 cycles. A. Readplots with top matching events (#1-#7) across samples highlighted in different colors. Boxes include information on read name, homology value, and the presence of untemplated inserts. **B.** Manual sequence alignments of reads covering the CSR junctions of clones #1 obtained after 15, 20 and 25 cycles

To address the impact of cycle number on SWIBRID output in more diverse sample setting, we performed an experiment with 200.000 PBMCs of six different donors analyzed after 5, 10, 15, 20 and 25 cycles of PCR amplification. Samples with 5 to 15 cycles did not pass our regular quality threshold of 500 filtered reads. 25 cycles yielded enough reads for reliable clustering and was consequently set as standard.

To compare reads and their represented clones within the same donor, we devised a meta-clustering strategy matching and tracing similar events in a single donor after low versus high cycle number amplifications (see Supplementary Note 2 and Figure X2). Thereby we identified 7 events with increased probability to represent B cells deriving from the same common ancestor (Figure X2A). The manual analysis of potential sister clones revealed that clone #1 was detected after 15, 20, and 25 cycles, as it represented with high sequence homology and matching breakpoint coordinates (51 nt of homology of sequences: SM chr14: 105860040-105860090 and SA1; chr14: 105711609-105712367) (Figure X1B). We have now added a Supplementary Note 2 that described how we estimate reproducibility by cluster tracing.

Importantly, it must be considered that 200.000 PBMCs with approximately 10.000 B cells and 5.000 switched cells will contain only minor overlap of clones with shared common ancestors.

The effect of cell number on SWIBRID output is addressed in main Figure 1e and results section line 145-156. When titrating the number of cells further, we observed that at 500 input memory B cells, the diversity does not reach 500 clusters. Such low diversity would be at risks of leading to unreliable measurements of SWIBRID features that depend on event averaging.

3. If the authors take the PBMC from one patient and divide in half and the PCR, and sequence, please show how similar or different the results are. For example, can they find the top 10 largest and top 10 smallest deletions in each of the two halves of this analysis?

The reviewer raises an important concern, pointing to the comparability of two replicates from the same donor regarding their outcome. As shown above in Figure X2, we indeed find shared clones across samples deriving from the same individual with certain clones present in independent triplicates (we have added the tracing plots in the new Supplementary Note 2).

We also analyzed how many clusters are shared when comparing replicates of 50,000 and 100,000 of IgG⁺IgA⁺CD27⁺ B cells (see now Figure 1f in the paper). When using purified class switched B cells as an input (increasing the likelihood to find shared clonotypes), samples clearly shared more clusters among replicates of the same donor (A-A, B-B and C-C), than among different donors (mixed). More quantitatively, we observed up to 26% shared clones in duplicate samples of 50,000 memory B cells from the same donor, while less than 20% of clusters were shared between samples from different donors (Figure X3 and new Figure 1f in the paper). The baseline likely reflects both, the limited resolution of our clustering methods as well as the possible presence of “public” clones expected from preferential switching in certain genomic loci. Shared VDJ heavy chain clonotypes have been estimated to range between 1% and 6% between two subjects (Soto, C., Bombardi, R.G., Branchizio, A. et al. Nature 566, 398–402 2019;

<https://www.nature.com/articles/s41586-019-0934-8>). However, no comparable studies providing estimates for CSR clonotypes are currently available.

Figure X3. Cluster tracing between donors from the cell number experiment, now Figure 1f. 50,000 (50k) and 100,000 (100k) CD27⁺IgA⁺IgG⁺ B cells were analyzed in replicates of donor A, B and C. Shown are % of shared clusters in replicates of donor A (A-A), replicates of donor B (B-B), and replicates of donor C (C-C) or from different donors (“mixed”)

4. Beyond the one or a few actual sequence junction examples shown, please provide 20 actual human patient junctions and the sequence features of each of these (from the pooled PBMC samples). I request this from all bioinformatic DNA sequence studies.

We thank the reviewer for this request and developed visualizations that show selected CSR junctions in the context of the read clustering, together with information on homologous or untemplated nucleotides, which are now integrated in the Supplementary Note 2. Besides the visualizations of the cell number experiment (see point 3 above), we also show a read clustering and junctions from random reads in the top 15 shared clusters from one healthy human donor, for which we sequenced amplicons both using minION and pacBio technology. Nucleotide homology and untemplated inserts are indicated in every read. This shows that the two methods give extremely similar results.

5. Because Ig CSR sequences are very repetitive, assessment of microhomology and perhaps even some other features may be difficult.

We agree with the reviewer on this important point. There are two issues to address: i) microhomology might be falsely interpreted as repair-associated homology, while it derives from the repetitive nature of the genomic loci, and ii) the applied methodology (including sequencing technology) hampers accurate assessment of microhomology.

Regarding the first point, it is true that microhomology may be misidentified due to the repetitive nature of switch region sequences. However, our intention is not to compare microhomology in CSR junctions with that observed in other, non-repetitive genomic regions. Instead, we aim to compare microhomology within CSR junctions across different donor contexts, specifically, in relation to varying defects in DNA repair and immune function.

To increase robustness and mitigate bias introduced by the methodology, including sequencing errors, most of our features are derived by averaging many reads and often also by binning events along the genome. The variance partitioning analysis in Figure 2b presents our approach to systematically attribute the “noisiness” of each feature to “relevant” factors such as donor identity or likely technical artefacts such as batch or sequencing method. We are therefore confident that the robust features we selected on an independent cohort of healthy donors can be quantified with sufficient accuracy.

More specifically, we included some nucleotide-level features such as homology and untemplated nucleotides around CSR junctions or blunt ends if neither is observed (see the annotated read plots now shown in Supplementary Note 2). Since switch regions are not directly comparable to less repetitive genomic regions, we at least aimed to relate our findings to other studies that specifically analyzed switch regions:

- Vincendeau et al.: SM-SG1 junctions of mouse splenocytes amplified by classical PCR and sequenced via pacBio
- Panchakshari et al.: CH12 mouse cell lines and mouse spleen samples enriched by linear amplification (HTGTS) and sequenced by Illumina short reads
- This study (SWIBRID): random human donors sequenced by minION and pacBio.
- This study (SWIBRID): mouse CH12 cell lines, spleens, and lymph nodes sequenced by minION.

To avoid adding more technical detail to the paper, we prefer to show the following comparison of methods to sequence CSR junctions just in this response.

Figure X4: Comparison of sequencing error rate, average number of homologous nucleotides around junctions, homology score (derived from binned breakpoint positions), average number of untemplated inserts around junctions, percentage of blunt breaks, or average sequence complexity around donor breakpoint positions among the different datasets: Vincendeau et al. (mouse/pacBio), Panchakshari et al. (HTGTS) and this study (minION).

Globally, we observe that despite very different effective sequencing error rates (accounting not only for PCR or sequencing errors but also for mappability), our nucleotide-based estimate of homology in mouse samples is comparable to those reported in other mouse studies. In contrast,

human samples show a marked divergence from the mouse data for nucleotide-based estimates of homology, while this difference was less prominent for the binning-based homology score (Figure X4).

In contrast to homology, we observed a markedly higher frequency of untemplated inserts and a lower proportion of blunt ends in our mouse data compared to the findings of Vincendeau et al. and Panchakshari et al. We assume that this discrepancy stems from methodological differences, specifically, whether the approaches used were capable of retrieving reads that span junctions in high- or low-complexity genomic regions (highly diverse or highly repetitive sequences). The extent to which such complex regions were captured varied considerably depending on the method applied (see Figure X5). We observed that the coverage of repetitive (low complexity) regions is influenced by primer design and methodology more than by sequencing technology. For instance, HTGTS junctions are biased towards the non-repetitive (high-complexity) parts of Sm and Sa, while SWIBRID allows to capture more junctions that originate from repetitive (low-complexity) regions (Figure X5).

Figure X5: Sequence complexity (top) and breakpoint frequency (bottom) for the human (left) or mouse (right) IgH locus, using different datasets of healthy human donors with pacBio or minION sequencing, or WT mouse spleens, lymph nodes, or CH12 cells with minION sequencing, mouse splenocyte data with pacBio sequencing from Vincendeau et al., or mouse CH12 or spleen data with short-read HTGTS sequencing. Junctions originating in low-complexity regions (red ellipses) in Sm or Sa are preferentially captured with our minION reads.

6. The use of the term ‘scar’ is inappropriate at all places in this manuscript. These are simply genomic alterations. Near the very end of PMID 36587186, that author explains the inception of the phrase ‘information scar’, which was not to be confused with any physical ‘scar’ in the DNA duplex structure. Since that inception, the dramatic use of the term ‘scar’ even for simple mutations has become disturbingly frequent in the literature, even though this is inappropriate, especially for non-coding regions such as the Ig CSR regions which

do not encode information. I provide this comment for all papers that I review that inappropriately use the phrase ‘genomic scar’.

We appreciate the reviewer’s comment and have replaced the term “scar” with “CSR junction” throughout the manuscript. We also changed the title accordingly “Recombination junctions from antibody isotype switching classify immune and DNA repair dysfunction”.

7. In the Supplementary file, lines 25-26, the authors write: ‘Given a high rate of sequencing errors, only the coverage of a position (not nucleotide identity) was considered, and gaps smaller than 75nt were ignored.’ Given the repetitive nature of Ig CSR sequences, could the authors explain this to me a bit more in their responses?

As shown in pages 12-13 of the Supplementary Material, we observe for minION but not pacBio that sequencing leads to an overrepresentation of small indels. This is even observed in a plasmid sample that did not undergo PCR amplification (Extended Data Figure 1g). When assigning different reads to a cluster, we therefore aimed to remove this kind of technical noise from the data, and opted for an approach where we only consider whether a certain genomic position is covered by a given read. We thereby ignore sequencing errors (expected in every ~30th nucleotide, see Figure X4 above). To further increase robustness and group reads with the same CSR junctions regardless of sequencing-induced indels, we removed small coverage gaps < 75nt, without ignoring larger structural rearrangements expected to be induced by switching. Removing small gaps is not strictly necessary for successful clustering, but it results in clusters that are much more clearly defined (i.e., reads in the same cluster are more similar, reads in different clusters are more distinct). We have expanded this important point in the Supplementary Note 1. *“Further, long-read sequencing is prone to create indels. Insertions are ignored in the pseudo multiple sequence alignment but are taken into account at other points in the pipeline if they are either templated and detected by mapping, or untemplated and occurring around larger break points, in which case they are identified in the breakpoint realignment step. We noticed that deletions of up to 75nt length appear frequently even in a non-PCR plasmid control, but much less often when using pacBio instead of minION sequencing technology. To increase the robustness of the clustering by preferentially comparing the positions of larger breaks rather than likely random small deletions, we ignored all gaps smaller than 75nt.”*

8. For CVID, it is my understanding that this has diverse causes that do not affect the actual DNA events of VDJ recombination or class switch recombination. Perhaps in the Supplement, the authors could comment on how or why the analysis gives a readout beyond simply the number of B cells that have an antigen receptor.

The main determinant of B cell phenotypes in CVID is the defect in maturation and terminal differentiation of B cells. While the majority of CVID patients exhibit normal VDJ recombination and retain mature naïve B cells, memory B cells, especially switched memory B cells, are frequently reduced.

Importantly, it is not only the number of switched memory B cells that is affected in CVID. Mutations in receptors involved in B cell activation, B-T cell interaction, and cytokine signaling can

significantly impact isotype composition, beyond simple numerical changes in B cell subsets. For instance, certain CVID patients show a more pronounced IgA deficiency than IgG, a pattern often linked to mutations in TACI (A. E. J. Poodt et al., *Clin. Exp. Immunol.* 156, 35–39 2009; <https://doi.org/10.1111/j.1365-2249.2008.03863.x>), while in contrast, BAFF-R mutations can present with normal IgA but reduced IgG levels (K. Warnatz et al., *Proc. Natl. Acad. Sci.* 106, 13945–13950 2009; <https://www.pnas.org/doi/10.1073/pnas.0903543106>).

Furthermore, a subset of CVID cases is associated with defects in DNA repair pathways. While most CVID patients do not present with classic monogenic DNA repair disorders, recent research indicates that such defects can contribute to CVID pathogenesis in selected individuals. Targeted sequencing studies have identified rare, likely pathogenic variants in key DNA repair genes, including ATM (ataxia-telangiectasia mutated), and components of the DNA double-strand break (DSB) repair machinery such as RAD50, NBS1, as well as mismatch repair proteins like MSH2 and MLH1 (C. E. Hargreaves et al., *J. Clin. Immunol.* 41, 1315–1330 2021; <https://pmc.ncbi.nlm.nih.gov/articles/PMC8310859/>, S. M. Offer, Q. Pan-Hammarström, L. Hammarström, R. S. Harris, *PLoS ONE.* 5, e12260 2010; <https://doi.org/10.1371/journal.pone.0012260>). For some of those factors such as MLH1, defects are expected to affect the sequence context surrounding CSR junctions (R. Chahwan et al., *J. Exp. Med.* 209, 671–678 2012; <https://pmc.ncbi.nlm.nih.gov/articles/PMC3328365/>).

In conclusion, with SWIBRID, we do not exclusively assess the absolute number of B cells in CVID but subtle changes in isotype composition and DNA-repair deficiencies that specifically impact CSR. However, lower B cell numbers translate into fewer reads and fewer clusters, which in turn affect diversity metrics as well as other features that are standardized to the number of reads or clusters.

As we considered it crucial to explain this point to our readers, we have now added an explanation in the discussion section of the main text line 383-391.

Again, I commend the authors for their impressive amount of work.

We thank the reviewer once again for the appreciation of our work.

Reviewer #2 (Remarks to the Author):

The manuscript by Vázquez García, Obermayer et al., presents a novel method named SWIBRID (SWItch junction Breakpoint Repertoire IDentification) designed to characterize DNA double-strand break (DSB) repair outcomes during antibody class switch recombination (CSR). Leveraging decades of foundational research into B-cell mediated DNA repair, this method uniquely profiles genomic scars resulting from CSR, allowing for assessment of immune diversity and DNA repair functionality from peripheral blood samples. Employing long-read sequencing and sophisticated bioinformatics pipelines, SWIBRID identifies distinct repair signatures in various DNA repair mutants and in patients with immunodeficiencies such as Common Variable Immunodeficiency (CVID) and cancer-associated DNA repair deficiencies. The authors demonstrate discriminatory power in distinguishing repair-deficient genotypes and patient cohorts, underscoring potential clinical utility in personalized diagnostics and early detection of pathogenic defects. Some issues still requiring clarifications include:

We thank the reviewer for recognizing our technology as a powerful new tool and for the thoughtful comments, which we found highly valuable, particularly in guiding us to strengthen the clinically relevant aspects of our work. We have carefully considered each point and addressed all comments in detail as outlined below.

1- The methodology shows promise in distinguishing specific DNA repair deficiencies and CVID. Have the authors considered potential confounding factors such as age-related changes in B-cell repertoire or treatment-related effects on CSR in clinical samples?

We thank the reviewer for this important comment. While we had shown that feature values do not correlate with age or gender (Figure 2c; original manuscript, lines 181–182), we had not assessed the potential influence of treatment. We have now included this analysis in main Figure 4B and in Extended Data Figure 4A. Treatments of patients are described in the methods section and Supplementary Table 2.

The vast majority of features were not correlated with drug treatment or immunoglobulin (Ig) supplementation. Treatments assessed were those given shortly before or at the time of sample collection and included: prednisone (n=3), cortisone (n=1), steroids (n=1), sirolimus (n=1), and infliximab (n=1). Ig supplementation was administered either subcutaneously (n=24) or intravenously (n=8). Overall, the feature values did not correlate with the drug treatment of Ig supplementation (Figure X6A), however, cluster entropy showed a difference between individuals receiving subcutaneous and intravenous Ig versus without treatment (Figure X6B).

It is important to note that this cohort is not optimal for specifically evaluating the effects of these treatments, as only one individual each received cortisone, steroids, sirolimus, or infliximab. For Ig supplementation, where a sufficient number of treated individuals was analyzed, we observed significant alterations in only two features of CSR junction patterns: the distribution of breaks across the SA1 and cluster entropy, an indicator of sample diversity.

However, it remains unclear whether these discrepancies are attributable to the treatment itself or to higher overall disease severity in these patients. Determining this would require analysis of samples obtained both before and after treatment, which was not possible for the donors included in this study.

Figure X6. Influence of treatment on SWIBRID results. **A.** Box plots of percent explained variance by donor, diagnosis, sex, age, batch, immunomodulatory treatment, or Ig administration for features in indicated categories. **B.** Box plots of cluster entropy values for CVID patients that received intravenous (ivig) or subcutaneous Ig (sclg) together with controls (black dots).

2- It remains unclear whether the CSR scar patterns detected by SWIBRID are stable over time in individual patients or influenced by transient immune activation events or infections. Are there any longitudinal patient data to draw such conclusions from in the analysed sample cohort?

While we did not systematically establish a longitudinal cohort at this point, we happen to have samples from the same two donors (one healthy: 09-0/1/2, one CVID-like: 32-0/1/2) at successive time points several months apart (09_healthy: 28.07.2020, 12.01.2016, 03.11.2020; 32_CVID-like: 23.03.2023, 02.10.2023, 11.05.2020). These samples cluster closely together in the PCAs (indicated by connecting lines) and heatmap of Figure 4, but also individual features are largely stable (Figure X7). Figure X7 depicts the heatmap of main Figure 4h with these 6 samples separated on the right.

We have now added a sentence in the main manuscript and expanded the legend of Figure 4 accordingly. “Although we did not establish a longitudinal cohort, samples from two individuals (numbered 09 and 32) collected over periods of up to four years clustered tightly in the heatmap and in PCA space (Fig 4e), indicating a stable SWIBRID profile over time.”

Important exceptions among the features that were less stable over time (e.g. in the CVID-like donor) relate to repertoire diversity, such as top clone occupancy or cluster entropy. While outcomes of DNA repair in an individual are generally expected to remain stable over time, exposure to pathogens (which can induce clonal bursts and cause temporarily skewed diversity) may lead to fluctuations in diversity related measures. We consider this variability a key advantage of the technology, as it also allows tracking changes over time, particularly relevant for detecting pathogenic clonal expansions associated with B cell lymphoma.

Figure X7: Heatmap of main Figure 4H with successive samples from the same two donors visually separated on the right. Replicates of donor 09: #1: 28.07.2020, #2: 12.01.2016, #3: 03.11.2020. Replicates of donor 32: #1 23.03.2023, #2 02.10.2023, #3: 11.05.2020.

3- While the method categorizes patients effectively, it would be highly beneficial to correlate specific CSR signatures with clinical outcomes or disease severity, if possible. Such correlation would substantiate clinical relevance and utility.

We thank the reviewer for this comment and fully agree that correlating SWIBRID outcomes with disease severity is of great value. However, the limited size and pronounced clinical heterogeneity of our current cohort make it challenging to address this question conclusively.

Among the CVID and CVID-like patients in our study, 23 are reported to have complications, and 10 are classified as “infection only”, of which two additionally have splenomegaly and one has diarrhea. Patients with complications represent a broad spectrum of immune dysregulation. Unfortunately, the complication data are not integrated in a way that allows us to assign a clearly defined severity score or value.

To address the question nevertheless, we evaluated whether SWIBRID feature-based patient classification correlates with the presence or absence of complications. While we did not observe a general rule, we noted that 15 out of 23 patients with complications appear in Cluster 3 (Figure X7). More specifically, 7 out of 9 patients with complex lung disease (Interstitial Lung Disease and Granulomatous-Lymphocytic Interstitial Lung Disease), and 8 out of 11 patients with more than one complication, fall into Cluster 3. This suggests that patients with more severe disease tend to cluster in Cluster 3 (71% of severe cases). While this trend is promising, we acknowledge that additional data will be needed to establish a definitive link between SWIBRID-based classification and disease severity.

In conclusion, to rigorously answer the reviewer’s question, a larger and clinically well-characterized cohort will be required. We have therefore decided not to comment on the correlation between SWIBRID features and disease severity in the main manuscript at this point.

4- The manuscript introduces several CSR-related genomic scar features. However, the biological mechanisms underlying some of these newly identified features (e.g., long templated insertions, certain sequence context motifs) require further experimental validation to clarify their roles in DNA repair pathway preference or dysfunction, or at least additional discussion to their origins and potential mechanisms.

We have addressed templated insertions in detail in our previous publication (*Lebedin et al., 2022*), where we showed that such insertions fall into two main categories: i) those derived from telomere-proximal regions and R-loops (45.4%), arising prior to the mature naïve B cell stage, and ii) those from early replication fragile sites (ERFS), appearing post-activation in memory B cells. Overall, 88% of inserts originated from mRNA-encoding regions, suggesting a link between transcriptional activity and insert origin. Even though this is a comprehensive study, yet another study would be necessary to identify specific molecular factors responsible for insertions. We reference our findings in our discussion “*Our previous studies*

showed that antibodies can gain an extra domain from large templated insertions in the S region, but the precise molecular factors contributing to these events remain elusive”.

We would also like to respectfully point out that this study was not designed to focus on a single molecular mechanism, such as templated insertions. Rather, our aim was to develop and validate a novel, *de novo* analytical framework that can be applied across mouse and human samples (including patient-derived material) ensuring both technical robustness and clinical applicability in patient contexts. We strongly agree with the reviewer that understanding the mechanisms behind specific features is a valuable future application of this technology. Our method provides a platform for studying DNA repair in human systems and may help reveal yet-unknown factors, as exemplified by recent discoveries like Shieldin and HMCES.

We nevertheless think that fully dissecting the underlying biology of individual CSR junction features would require a dedicated project, involving genetic models, perturbation studies, and functional validation, and is thus beyond the scope of this work.

5- The manuscript employs machine learning (ridge regression) to classify genotypes. Detailed clarification on model robustness, cross-validation methods, and generalizability to external datasets would be ideal.

We thank the reviewer for requesting more information on that important point and have expanded the corresponding parts in the results and methods sections line 743-745. Specifically, we separated our data into testing and training sets, constructed the regression model on the training data and evaluated on the test data. Additional cross-validation was used to get additional measures of accuracy.

As our data presents the first of its kind and most of our features are strongly correlated to isotype frequencies, we cannot apply our model on external published data generated with different methods, especially regarding primer design. We therefore used a different cohort of DNA repair deficient patients from a different hospital to test generalizability of the prediction model.

Minor Concerns:

1. Line 141–147: clarify units—reads or cells?

We apologize for this confusion. In lines 141-147 of the original manuscript we explain two different approaches to address the performance of our analysis pipeline. To test the impact of reads and cells on SWIBRID output, we modified the number of reads in the Simulated Reads data (main Figure 1f) and the number of cells in the cell number experiment (main Figure 1e). We have now modified the main text for clarification “*To further confirm an adequate quantification of CSR clones and understand the limitations of the pipeline, we assessed different numbers of i) in silico generated CSR-reads representing diverse clonotypes and ii) primary B cells to address the impact of the input number on output clusters.*”

2. Extended Data Fig. 1f,g legend missing information to be understandable.

Thank you for your comment. The Extended Data Figure 1 f–g presents data obtained from a linearized plasmid containing the human switch μ region as explained in the materials and methods section. Linearization of this plasmid via a restriction enzyme digest allowed us to analyze a sample in absence of PCR amplification to judge the impact of PCR on our analysis.

We have not modified our legend to clarify: “**f** a plasmid containing the human S_{μ} region was linearized via a restriction enzyme digest, representing an outcome in absence of PCR amplification. **g** Depicted is a read plot obtained from the linearized plasmid with 2,500 reads.”

3. Figure quality: many diagrams use colour-only encoding (e.g., PCA). Or have overlapping color legends with such as fig 2b.

We have aimed to use the same color palettes across the manuscript (coding for feature classes in Figure 2a, explanatory factors in Figure 2b, genotypes in Figure 2g and Figure 3c,d,e, tissues in Figure 3b). We agree that colors in Figure 2b can be confusing and have changed the palette.

4. Methods: provide accession for training datasets (mouse Vincendeau et al.)

Training datasets are available in the documentation of SWIBRID in the github repository (bihealth.github.io/swibrid). We have now indicated it more precisely in the manuscript in the methods section: “**Vincendeau et al. re-analysis**: Sequencing data of CSR junctions generated by Vincendeau et al. were downloaded from SRA (accession number PRJNA831666) and processed with standard SWIBRID settings for mouse, after adjusting coordinates of the relevant switch region for different primer locations.”

5. Some sections, particularly those presenting comparisons and statistical significances, could benefit from clearer methodological descriptions, including exact statistical tests used and justification for the chosen methods.

We have expanded the description of tests used including multiple testing correction in the captions of Figure 2 and Extended Data Figure 3. In addition, a more extended explanation of the variance partitioning analysis in the legends of Figure 2, the ridge regression in Figure 3-4, and the number of features used for the different PCAs (see point 5 above) are now added. A more detailed description of features used in the PCA can be found in the Supplementary Table 1.

6. The authors acknowledge technical noise from sequencing errors. More explicit discussion and mitigation strategies for these technical limitations, including error rates of Nanopore versus PacBio technologies, are essential for readers to interpret data quality accurately.

We are aware that minION sequencing has a high error rate in comparison with other sequencing technologies (Figure X4). In response to reviewer 1 (point 5), we compared the impact of the sequencing technologies minION and pacBio on our results. SWIBRID feature values are consistent across sequencing technology, as illustrated by the comparable nucleotide-based estimate of homology. In response to Reviewer 1 (point 4), we present read clustering and CSR junctions from random reads within the top 15 shared clusters of a healthy human donor, for whom amplicons were sequenced using both minION and pacBio technologies, demonstrating that the two methods yield highly similar results. We have added the tracing plots in the new Supplementary Note 2. Extended Data Figure 2g shows an explicit comparison, while Figure 2b, d and Extended Data Figure 2f show the influence of sequencing technology on features.

7. Discussion on how SWIBRID might compare to existing clinical tests (e.g., TREC/KREC assays) could provide valuable context for clinical integration.

We thank the reviewer for this comment and fully agree that comparing our method to existing TREC/KREC assays is relevant. While we do not consider SWIBRID to be an alternative to TREC/KREC, we regard it as a complementary diagnostic tool. TREC/KREC assays are well-established for assessing V(D)J recombination capacity, primarily used to identify severe immunodeficiencies characterized by profoundly reduced T and/or B cell numbers early in life.

In contrast, SWIBRID is designed to detect immune disorders that are not well captured by these conventional assays, such as Common Variable Immunodeficiency (CVID). Our method specifically enables the identification of defects in class switch recombination and the loss of class-switched memory B cells, which are key features of CVID that may not be apparent through TREC/KREC testing.

We have now addressed this point more explicitly in the discussion section, to better emphasize the distinct and complementary roles of SWIBRID and TREC/KREC in immunodiagnosics: *“Overall, SWIBRID enables the identification and characterization of class switch recombination defects, key features of CVID that may be missed by TREC/KREC testing.”*

In conclusion, the manuscript significantly advances the field by introducing a powerful new method for profiling DNA repair competency and immunological health through CSR scars. Addressing the highlighted concerns will enhance the robustness, clarity, and broader impact of this promising study.

We once again thank the reviewer for this assessment and appreciate the recognition of our work as a valuable methodological contribution.

Reviewer #3 (Remarks to the Author):

The authors of Garcia and Obermayer et al describe SWIBRID to identify the repertoire of switch junction breakpoints that uses PCR and long read sequencing from blood samples to detect DSB repair patterns driven by IGH class switch recombination (CSR) in the B cell subgroup. In surveying patient samples, the authors report immunodeficiencies, DNA repair defects, and patient-specific scarring signatures with high accuracy. The work looks to be well developed with many identified features that may be useful for sub-group distinction across human and mouse samples but this reviewer was struggling to figure out which new features were the most frequent and most relevant to the CSR repair outcome repertoire. The manuscript could be improved further by addressing the following comments below:

We thank the reviewer for acknowledging the dedicated effort that went into developing our technology and for the constructive criticism regarding the relevance of specific features for repertoire outcomes. We have carefully addressed all comments and believe they were very helpful to improve the manuscript.

Major Comments

1. Most of the 68 features in 5 categories are presented as fractional data, presumably to normalize for absolute variance measures. It would be very helpful to have all of the data outlined in the figures mirrored in absolute frequencies in supplemental tables; this would be analogous to showing the films of blots. It is important for readership to evaluate which features are weighted by high frequency, the rigor of the sequencing data, and to confirm any statistically significant shifts in proportional data are not influenced by low frequencies which may make some highlighted features too noisy for evaluation.

We agree that sharing of raw data is essential for evaluation of reproducibility, which is why our github repository (bihealth.github.io/swibrid) does not only contain the code needed to create the figures but also the raw output of the SWIBRID pipeline. Raw reads for mouse data are available on the sequence read archive, while we cannot currently share human read data due to patient privacy concerns.

We are not entirely certain what is meant by “absolute frequencies”, as most features in our analysis are obtained by first averaging over all reads within a cluster, and then over all clusters in a sample. Given that most samples contain more than 10,000 reads, we posit that most features are relatively robust unless they measure rare events like templated inserts. To systematically evaluate the noisiness and robustness of our features, we used the variance partitioning analysis of Figure 2b, which lets us quantify exactly what proportion of the observed variability is attributable to inter-individual differences rather than technical effects like batch or sequencing method. We therefore believe that the resulting 44 robust features are not strongly affected by low frequencies.

We have reformulated and expanded the corresponding section in the manuscript to better convey this point (“*To assess the robustness of our features across different batches or sequencing methods, and their ability to discriminate between donors, we used variance partitioning analysis to verify that for most features, the bulk of the variability is explained by donor identity (Fig. 2b,c, Extended Data Fig. 2f). This indicates that our quantification yields accurate and reliable output.*”)

2. Adding to the first comment, too many features will naturally yield some measurable difference that may not be related to biology unless the magnitude of difference withstands the increased number of multiple comparisons. As the central point of the manuscript is to highlight the numerous features, then proper statistical rigor needs to account for false discovery.

We thank the reviewer for raising this important point about statistical methodology. We would like to emphasize that our central aim is not the identification of specific “differential” features but rather of systematic differences in the breakpoint patterns between conditions. The features are just a way of capturing multiple aspects of these patterns in ways that we hope are somewhat intuitive. Accordingly, we are not primarily using multiple hypothesis testing and p-value adjustment to identify specific differential features, but rather dimensionality reduction (PCA) and regression with a training / testing split, which is where statistical rigor primarily derives from. Even if some features would show spurious significance in an un-adjusted test, we would not expect to see a systematic difference in a PCA or high performance of a regression model unless multiple features show coordinated effects.

Nevertheless, we have added a multiple testing comparison to Figure 2h and Extended Data Figure 3a (where many features are shown), while statistical comparisons of features shown in box plots (e.g., Figure 3d, 4c,j, Extended Data Figure 4b) are meant to highlight interesting or informative features for interpretation of the systematic group differences observed in PCA or regression, but not to justify differences between groups in the first place. We are therefore less concerned with false discovery here.

3. Fig2a is somewhat helpful for revealing the categories but a detailed description of the full 68 features should be outlined in a supplemental figure/table so readers can fully evaluate the data presented. The feature names are bulky. Some terms were not readily understood by this reviewer (i.e. not clear from only looking at the figures as a litmus test for comprehension) and will certainly alienate most readers. Please reconsider renaming for efficiency and clarity that could pair with detailed description of the filters/ranges used to define each feature. Fig2h and related would certainly benefit.

We thank the reviewer for this comment, as we aim to maximize the readability and clarity of our manuscript, including the use of self-explanatory feature names. We now renamed the features to make them easier to understand. For example, we have improved clarity by adding the primary measure or unit, such as percent (*pct_*) or mean (*mean_*), by improving the understanding of the

measure, such as distribution of breaks along the switch region (*break_dispersion*) and by naming first the most important aspect of the feature (*occupancy*). We adapted the text and the figures accordingly. Supplementary Table 1 has been updated and includes a description of each measure. Additional feature documentation is available at bihealth.github.io/swibrid accordingly.

4. Furthermore, for each experiment using the feature/category breakdown, a list of ranked features impacting the experiment should be provided. If none are impacting, then indicate none (presume this is the case for Spleen Lig4 in EDFig3b? If not need to indicate in figure legend why no Lig4 data for spleen). In this regard, it is difficult to follow the logic as to why 32 features were used for one set of knockouts and 39 features in another, what those features were and why others were excluded; it would appear that the remaining 20 or so have little value if they are not used frequently.

We thank the reviewer for this important remark.

We have now revised the legend of Figure 3 to clarify that knockouts of DNA repair factors were studied only in CH12 cells, whereas splenocytes and lymph nodes were derived from wild-type mice: *“Principal component analysis (PCA) of WT CH12 cells, CH12 cells with knockouts (KO) of Brca1, Lig4, Rif1, and Trp53bp1, as well as splenocytes, and lymph nodes from WT mice using robust SWIBRID features.”*

The features were not selected arbitrarily. First of all, we only use the most robust features as identified in Figure 2 through the analysis of human samples, defining a robust set of 44 features. Second, the captions mention the number of features used in each specific PCA. This number depended on the dataset: mouse vs. human, CH12 vs. splenic B cells, our data vs. Vincendeau, which all contain different information. As an example, CH12 switch exclusively to IgA, which excludes any features related to the IgG isotype. In contrast, Vincendeau and colleagues focused specifically on IgG1, using primers that exclude all other isotypes. As a result, the design of each dataset limited the number of usable features. We apologize that this was not clearly explained in the original manuscript and have now added a clarification to the corresponding figure legends of Figure 2 and 3.

“PCA of re-analyzed Vincendeau data³⁰ using 32 of 44 robust features, excluding features related to isotypes missing in the dataset and features missing in mouse compared to human.”

“Left: including reads from all switch regions (42 of 44 robust features - excluding features missing in mice compared to human). Right: selecting reads with Sa primer (33 of 44 robust features – excluding sg and human specific features).” Furthermore, the features used for every analysis are listed in Supplementary Table 1.

To address feature importance, we included the respective plots in Figure 3f, Extended Data Figure 4c and the new Extended Data Figure 4g (please see also our response to the next point).

Finally, Panchakshari et al. spleen data did not contain a Lig4 KO, which is why this comparison is missing (Figure 1 in Panchakshari et al.).

5. In terms of PCA, you need to indicate which components distinguish each of your comparisons and which features overlap. This is otherwise generically saying there are differences and similarities or they are separated from controls, which have no added value to the field given the prior work on the genes implicated or to provide causal sources for distinction in CVID/CVID-like samples; this latter part is partially done but needs to be applied to earlier parts.

Related to an earlier point raised by the reviewer, our main aim indeed is to identify fingerprints of specific DNA repair defects or immunodeficiencies as manifested in a CSR junction pattern. If such a fingerprint happens to be sufficiently well captured by individual features that easily lend themselves to mechanistic or causal interpretation, that is of course a highly desirable outcome. Nevertheless, our method provides a valuable biomarker even if that is not the case.

Each of our PCAs is accompanied by other plots that highlight distinctive features (e.g., Figure 2h for the PCA in 2g, Figure 3c,d,f for the PCA in Figure 3e, 4c,h,j and Extended Data Figure 4c for the PCAs in Figure 4d-f, Extended Data Figure 3a,c for the PCA of Extended Data Figure 3b).

We agree with the reviewer that for the patient samples this point was less clear. We therefore added another plot (Extended Data Figure 4g, which is also Figure X8) showing the weights for the regression model of Figure 4g to indicate which features are most useful in distinguishing CVID from healthy.

Figure X8. Top 19 coefficients in a multinomial ridge regression model derived from the 29 training samples. Features are colored according to the classification of Figure 2a. Higher coefficients indicate higher values in the CVID group.

6. Related to point #5 is that a laundry list of features are indicated but needs to be distilled down to what those robust affecting features mean in aggregate.

We appreciate the comment of the reviewer. In this case, we would point out that a high-throughput approach such as our presents a more unbiased and comprehensive view of CSR junction patterns than more targeted methods aimed at elucidating specific effects in specific conditions. We agree that the resulting data is highly complex and therefore hard to summarize in an intuitive way. That is why the features were designed by leveraging what has been used in previous studies, e.g., homology, plus, some other features that capture not yet recognized aspects in this region or in this context, e.g., break dispersion. Importantly, not all of these features are easily interpretable or directly and causally linked to a specific immunodeficiency or DNA repair defect, even though many of them show systematic and very specific differences between different mouse genotypes or the different human patient groups we studied. While our phenomenological approach for now is more similar to other biomarkers of immune health that

integrate disparate datasets (e.g., R. Sparks et al., Nat. Med. 30, 2461–2472 2024; <https://www.nature.com/articles/s41591-024-03092-6>), we expect that future cohorts allow to refine and extend our observations.

7. Methods section does not describe machine learning and more detail is needed beyond multinomial ridge regression for each experiment they were used. For independent reproduction of findings, the specific features used for each prediction experimental sets should be listed and any additional values, range for confidence, number of technical versus biological replications, etc. in the model should be reported. If this software is not cited in MS, please make sure this is included.

We understand the reviewer's concern and have expanded the methods section to more clearly state that we use standard R models as well as their inputs: "*For the CH12 dataset, we used 33 features plus 49 samples for training and 36 samples for testing. For the CVID dataset, we used 44 features plus 29 samples for training, 26 samples for testing, and 21 samples for validation.*". In the manuscript, details about the machine learning models can be found in: i) Figure. 3a and 4a summarizing the inputs used for our machine learning models and ii) the repository at github.com/bihealth/swibrid_paper containing all code and data needed to reproduce the figures.

Minor Comments

8. This manuscript suffers from poorly framed figures with text and symbols that are either too crammed together, too small, or using inappropriate axis labeling. For the latter, there are no such fractional n_homology measures and listing 0.5 increments only serves to cause confusion to the lay reader. Labeling format (e.g. n_homology, Fraction IGA VDJ) may be understandable as a programming label for the authors but is not for lay audience. This reviewer urges the authors to carefully consider the context of the data shown and reformat the figure panels to make it more conceptually available to a broader audience.

We understand the concern of the reviewer and have worked to generate clearer figures. First, we renamed features in pursue of a clearer definition and we now leave more space in the figures. The updated names of the features can be found in Supplementary Table 1 in the particular case of n_homology, we changed it to "homology", although it is calculated as an average over the entire sample, leading to fractional values. In the rest of the cases, we have added "pct" for percentage, instead of "frac" for fraction to ease the understanding of the features.

9. Not clear what type of V(D)J sequencing was occurring in the study as it is noted in the manuscript. Fig 1F labeling is also not clear. Was any variable region analysis performed and how was that integrated into SWBRID comparison? Line 152 is related to the same

panel but in comparison to IGHA vs IGHG. There seems to be a disconnect in the text and figure.

In the manuscript, we attempt to compare SWIBRID with the most standard method to analyze B cells, which in this case is BCR repertoire sequencing. Beside information on VDJ we can also retrieve the information of the respective isotype. To understand whether our genomic approach is reflected in the RNA, namely CSR junction alleles matching BCR-mRNA isotypes, we chose to compare the frequencies of IGHA vs IGHG isotypes in SWIBRID versus VDJ-sequencing datasets. We have now modified the respective sentence to make this clearer: *“To compare SWIBRID with the standard approach for B cell analysis and evaluate whether our genomic findings are reflected at the RNA level, we analyzed IGHA and IGHG isotype frequencies between CSR junction alleles and BCR mRNAs from circulating B cells. [...] Both approaches obtained similar inter-donor differences in IGHA fraction”*

As now explained in the methods section, the VDJ library is generated by template switch cDNA synthesis, followed by the amplification of BCR transcripts using a mixture of constant primers and template switch sequence. Illumina sequencing was then performed and the BCR transcript analysis with MiXCR using the “generic-bcr-amplicon-umi” preset with the tag pattern `"^N{22}(UMI:N{16})N{7}(R1:*)^(R2:*)"`.

10. There is a disconnect in feature numbers for iii on line 163 versus Fig 2a. Please clarify.

We thank the reviewer for identifying this error. We have corrected the information in Figure 2a. It is only 28 features and we have 2 more breakpoint dispersions for mouse Sg2b and Sg2c.

11. Lines 171-172. Presuming only data from 1 of the 10 batches are relevant given that was the only batch using long read sequencing, please make sure to break down that information when comparing this to C2. If using both sequencing formats, then indicate as such for both cohorts. Only 6 of the 68 features are referenced; why those 6? Are they representing the majority of the data? If not, then why only show a minority? Brief explanation in the main text or more detailed explanation in the Supp. Note may be useful.

We apologize if there is confusion about the sequencing method we used. All our data is generated using long-read sequencing, mostly with minION and one batch of 10 samples in C1 with minION and pacBio, as stated in line 171, now line 176-177. The variance partitioning allows to isolate the amount of variability attributable to sequencing method even if not all batches use both sequencing methods. Figure 2d shows 6 exemplary features that we thought were useful to illustrate value ranges, while Extended Data Figure 2g shows agreement between sequencing technologies for all 44 robust features. The new supplementary note 2 also shows a direct comparison of sequencing methods for individual CSR junctions.

12. PCAs need to have indicated in legend how many robust features were ultimately used since this seems to change across experiments.

The legends now mention how features were used as well as Supplementary Table 1. Please see also response point 4.

For Fig2h and EDFig3a graphs you are better off showing only those with FDR-adjusted significance as showing everything despite some with larger symbols is uninterpretable and providing too much unnecessary information.

We now increased the contrast between significant (after adjustment) and non-significant features and added a multiple testing correction; however even non-significant features contribute to the reported correlation and are therefore not irrelevant.

Lines 260-261: not clear whether this new observation of GC content on the acceptor side of the old data that is not shown is conserved in your experiential data on Lig4 deficiency in the prior paragraph.

We apologize for the confusion and have added some details in the main text to improve readability. The GC content on the acceptor side (acceptor_score_W) is shown in Extended Data Figure 3c. The data on Ligase 4 KO in the HTGTS data aligns with the data obtained with CH12, as shown in the heatmap (Figure 3c).

Lines 287-288: Can the authors speculate in the discussion as to what the correlation means

We prefer not to speculate on this correlation. There is currently no available literature to help interpret this finding. Additionally, our cohort is clinically highly diverse, comprising patients with a wide range of complications and a unique spectrum of heterogeneous disease etiologies.

V(D)J is more accurate than VDJ and symbolizes that some antigen receptor loci (i.e. lambda, kappa, alpha, gamma) only use V-J recombination.

We agree with the reviewer that V(D)J is more appropriate when referring to BCR repertoires and in all cases when both heavy and light chains are considered. However, in our study we analyzed only the heavy chain sequences, all of which include the D element. Therefore, the use of VDJ is correct in our specific context.

- **Reviewer #1 (Remarks to the Author):**

My goal is to continue to support publication. I respect the authors responses to my questions. Importantly, the authors are applying long-read sequencing to a repetitive, GC-rich region, as they acknowledge. So, their experience with difficulties is part of the VALUE of this paper. I strongly suggest (almost insist) that the authors include in their Supplement the figure that they provided in their response to me (Fig. X1), along with the text (and legend) they wrote associated with X1: "the number of PCR cycles was a critical parameter requiring careful optimization. On the one hand, we needed sufficient amplification to enrich CSR junctions and generate enough material for the minION library preparation; on the other hand, we aimed to keep the cycle number as low as possible to maintain diversity. When PCR is performed in regions of repetitive DNA, recombination can create artificial diversity, which in a monoclonal cell line would appear as heterogeneous outcomes. We do observe some diversity after 25 PCR cycles (Figure X1A-B). However, our read plot (Figure X1A) and sequence alignment of 8 random reads (Figure X1C) still show considerable homology in a monoclonal sample with two switched alleles, including i) highly similar read alignments to the switch regions (Figure X1A) and ii) substantial nucleotide homology in the alignments (Figure X1C), enabling precise breakpoint identification. As PCR-induced artifacts increase with cycle number, we conclude that 25 cycles under our optimized conditions introduce only minimal artificial diversity."

I think that the diversity in the black histogram peaks that is clear in Figure X1 (panel A, left upper edge) seems to indicate to my naive mind that the amplification of this monoclonal cell line gives heterogeneous switch junctions at each of the two alleles, especially the upper one. Or do these authors have a different interpretation of this? Their comments on this are valuable regardless of their response. They should rest assured that I am not going to use their response against them. But readers, along with me, deserve clarity on this point, and I think these excellent authors also wish to be clear as well.

Response

We appreciate the reviewer comment and support to make the manuscript more comprehensive. We fully agree with the particular challenges that long-read sequencing of a repetitive, GC-rich region presents. We emphasized this struggle adding a sentence for clarification "The number of PCR cycles (25) was kept as low as possible to minimize technical noise while still ensuring sufficient representation of the biological diversity" (lines 118–119).

Reviewer 1 also raised a remaining concern regarding heterogeneity in switch junctions within our monoclonal cell line. We also agree with the concern towards the CSR junction heterogeneity in the monoclonal cell line (Fig. 1b). In the revised manuscript, we added the following statement together with read alignments in Extended Data Fig. 1h "We observed some heterogeneity in the monoclonal cell line, where subtle sequence variations around the breakpoint (Extended Data Fig. 1h) formed independent clusters in the read plot, leading to a slight overestimation of diversity for this sample." (lines 131-133)

Importantly, our results do not suggest that PCR is the main driver of this heterogeneity. To make this clearer, we have included our interpretation in the text "PCR may introduce such sequence variations, but a similar level of heterogeneity in both PCR-amplified samples and the non-PCR plasmid control suggests that most of the variation arises from clustering, mapping, and sequencing errors." (lines 133-136)

- **Reviewer #2 (Remarks to the Author):**

The authors have satisfactorily addressed my concerns.

Thank you for the comment and the support on the manuscript.

- **Reviewer #3 (Remarks to the Author):**

The authors made substantial improvements throughout the manuscript. This reviewer has no additional constructive comments to add.

Thank you for the comment and the support on the manuscript.

Reviewer #1 (Remarks to the Author):

The authors seem to have responded to my comments, though in a somewhat minimal manner. I am ok with it.

We regret that the reviewer considers our reply minimal. To our understanding, all aspects of the respective point-by-point comment have been incorporated into the revised main text (see colour marked text below). The elements were distributed throughout the manuscript to preserve its flow. Only the inclusion of the exact Fig. 1X was not met, as it would reproduce elements from the main figures and create redundancies.

Regarding PCR diversity, we still share the reviewer's view that it is a general matter of concern. However, as we observed similar heterogeneity in a PCR- and a non-PCR-amplified sample, we decided not to discuss this aspect in greater detail.

Point-to-Point:

The number of PCR cycles was a critical parameter requiring careful optimization. On the one hand, we needed sufficient amplification to enrich CSR junctions and generate enough material for the minION library preparation; on the other hand, we aimed to keep the cycle number as low as possible to maintain diversity. When PCR is performed in regions of repetitive DNA, recombination can create artificial diversity, which in a monoclonal cell line would appear as heterogeneous outcomes. We do observe some diversity after 25 PCR cycles (Figure X1A-B). However, our read plot (Figure X1A) and sequence alignment of 8 random reads (Figure X1C) still show considerable homology in a monoclonal sample with two switched alleles, including i) highly similar read alignments to the switch regions (Figure X1A) and ii) substantial nucleotide homology in the alignments (Figure X1C), enabling precise breakpoint identification. As PCR-induced artifacts increase with cycle number, we conclude that 25 cycles under our optimized conditions introduce only minimal artificial diversity.

Manuscript:

Lines 118-119: The number of PCR cycles (25) was kept as low as possible to minimize technical noise while still ensuring sufficient representation of the biological diversity.

Line 133-134: PCR may introduce such sequence variations specially in highly repetitive DNA

Line 129-133: As expected, analysis of an in-house monoclonal cell line revealed two main CSR clones in accordance with 2 switched alleles (Fig. 1b, c). We observed some heterogeneity in the monoclonal cell line, where subtle sequence variations around the breakpoint (Extended Data Fig. 1h) formed independent clusters in the read plot, leading to a slight overestimation of diversity for this sample.